# Femtosecond formation dynamics of the spin Seebeck effect revealed by terahertz spectroscopy

Tom S. Seifert[1,2], Samridh Jaiswal[3,4], Joseph Barker [5], Sebastian T. Weber[6], Ilya Razdolski[1], Joel Cramer[3], Oliver Gueckstock[1], Sebastian F. Maehrlein[1], Lukas Nadvornik[1,2], Shun Watanabe[7], Chiara Ciccarelli [8], Alexey Melnikov[1,9], Gerhard Jakob[3], Markus Münzenberg[10], Sebastian T.B. Goennenwein [11], Georg Woltersdorf[9], Baerbel Rethfeld[6], Piet W. Brouwer[2], Martin Wolf[1], Mathias Kläui [3] & Tobias Kampfrath[1,2]

Understanding the transfer of spin angular momentum is essential in modern magnetism research. A model case is the generation of magnons in magnetic insulators by heating an adjacent metal film. Here, we reveal the initial steps of this spin Seebeck effect with <27 fs time resolution using terahertz spectroscopy on bilayers of ferrimagnetic yttrium iron garnet and platinum. Upon exciting the metal with an infrared laser pulse, a spin Seebeck current $j_s$ arises on the same ~100 fs time scale on which the metal electrons thermalize. This observation highlights that efficient spin transfer critically relies on carrier multiplication and is driven by conduction electrons scattering off the metal–insulator interface. Analytical modeling shows that the electrons' dynamics are almost instantaneously imprinted onto $j_s$ because their spins have a correlation time of only ~4 fs and deflect the ferrimagnetic moments without inertia. Applications in material characterization, interface probing, spin-noise spectroscopy and terahertz spin pumping emerge.

[1] Department of Physical Chemistry, Fritz Haber Institute of the Max Planck Society, 14195 Berlin, Germany. [2] Department of Physics, Freie Universität Berlin, 14195 Berlin, Germany. [3] Institute of Physics, Johannes Gutenberg University Mainz, 55099 Mainz, Germany. [4] Singulus Technologies AG, 63796 Kahl am Main, Germany. [5] Institute for Materials Research, Tohoku University, Sendai 980-8577, Japan. [6] Department of Physics and Research Center OPTIMAS, Technische Universität Kaiserslautern, 67663 Kaiserslautern, Germany. [7] Department of Advanced Materials Science, School of Frontier Sciences, University of Tokyo, Chiba 277-8561, Japan. [8] Cavendish Laboratory, University of Cambridge, Cambridge CB3 0HE, UK. [9] Institute of Physics, Martin-Luther-Universität Halle, 06120 Halle, Germany. [10] Institut für Physik, Universität Greifswald, 17489 Greifswald, Germany. [11] Institut für Festkörper- und Materialphysik, Technische Universität Dresden, 01062 Dresden, Germany. Correspondence and requests for materials should be addressed to T.K. (email: tobias.kampfrath@fu-berlin.de)

Transfer of spin angular momentum between two subsystems is a common process in modern magnetism research and highly relevant for the implementation of spintronic functionalities[1]. In contrast to electrical currents, spin transfer can be induced not only by the flow of conduction electrons, but also by torques exerted between the subsystems[2,3]. A model case of incoherent spin torque is the spin Seebeck effect[2-7] (SSE), which is typically observed at the interface[3,8,9] of a magnetic insulator (F) and a nonmagnetic metal (N) (see Fig. 1). By applying a temperature difference $T^N - T^F$, a spin current with density[2,10]

$$j_s = \mathcal{K} \cdot \left( T^N - T^F \right) \tag{1}$$

across the interface is induced where $\mathcal{K}$ is the SSE coefficient. Since F is insulating and N nonmagnetic, $j_s$ is carried by magnons in F and by conduction electrons in N. It is readily measured in the N layer through the inverse spin Hall effect (ISHE), which converts the longitudinal $j_s$ into a detectable transverse charge current $j_c$ (Fig. 1). In the case of temperature gradients in the F bulk, magnon accumulation at the F|N interface can make an additional contribution to Eq. (1)[3,11].

Note that Eq. (1) presumes a static temperature difference and a frequency-independent SSE coefficient $\mathcal{K}$. It is still an open question how the SSE current $j_s$ evolves for fast temperature variations and in the presence of nonthermal states. Insights into these points are crucial to reveal the role of elementary processes in the formation of the SSE current, for instance magnon creation[12] in F and spin relaxation[13] in N. The high-frequency behavior of the SSE is also relevant for applications, such as magnetization control by terahertz (THz) spin currents[14-16] and spintronic THz-radiation sources[17-21].

In previous time-resolved SSE works, a transient temperature difference $T^N - T^F$ was induced by heating the N layer with an optical or microwave pulse[8,9,22,23]. It was shown that Eq. (1) remains valid on the time resolution of these experiments, from microseconds[8] through to ~0.1 ns (ref. [9]) and even down to 1.2 ps (ref. [23]). To search for the SSE speed limit, even finer time resolution is required, ultimately reaching the 10 fs scale, which resolves the fastest spin dynamics in magnetic materials[24].

In this work, we reveal the initial elementary steps of the longitudinal SSE by pushing its measurement to the THz regime. Upon exciting the metal of a prototypical F|N bilayer structure with an infrared laser pulse, the dynamics of the spin Seebeck current $j_s$ versus time $t$ are determined with a resolution better than 27 fs using the ISHE and electrooptic sampling. We find that $j_s(t)$ rises and decays on time scales of ~100 fs. The decay directly follows the cooling dynamics of the N electrons as seen in the transient sample reflectance. An analytical model shows that $j_s(t)$ monitors the density of the transient electrons and holes in the metal quasi-instantaneously because their spins have a correlation time of only ~4 fs and deflect the ferrimagnetic moments without inertia. Simulations consistently reveal that the rise of $j_s(t)$ mirrors the thermalization process during which the photoexcited electrons approach a Fermi–Dirac distribution. This observation highlights that efficient spin transfer critically relies on carrier multiplication and is driven by conduction electrons scattering off the metal–insulator interface. Our results are relevant for a large variety of optically driven spin-transfer processes. Applications in material characterization, interface probing, spin-noise spectroscopy and THz spin pumping come into reach.

## Results

**Experiment**. A schematic of our experimental setup is shown in Fig. 1 and detailed in the Methods section. In brief, we use ultrashort laser pulses (duration 10 fs, center photon energy 1.6 eV, pulse energy 3.2 nJ) from a Ti:sapphire laser oscillator (repetition rate 80 MHz) to excite the metal of yttrium iron garnet (YIG)|platinum (Pt) bilayers. Any spin current $j_s(t)$ arising in the metal is expected to be converted into a charge current $j_c(t)$ by the ISHE with a bandwidth extending into the THz range[17].

These extremely high frequencies are, however, inaccessible to electrical measurement schemes. We, therefore, sample the transient electric field of the concomitantly emitted electromagnetic pulse by contact-free electrooptic detection over a bandwidth of 45 THz. This technique allows us to determine the spin current $j_s(t)$ with a time resolution better than 27 fs (see Methods and ref.[25]). To monitor the electron dynamics in the Pt thin film, we also measure its transient reflectance by a time-delayed optical probe pulse (see Fig. 1).

**THz emission from YIG|Pt**. Typical THz electrooptic signals $S$ versus time $t$ for a YIG(3 μm)|Pt(5.5 nm) bilayer are displayed in Fig. 2a. The signal inverts when the in-plane sample magnetization **M** is reversed. Since the SSE current is expected to be odd in **M**, we focus on the THz-signal difference $S_- = S(+\mathbf{M}) - S(-\mathbf{M})$ in the following.

Figure 2b shows the amplitude of $S_-(t)$ as a function of the external magnetic field, along with the sample magnetization **M** measured by the magnetooptic Faraday effect (see Methods). First, both curves coincide. Second, the THz electric field associated with $S_-(t)$ is found to be linearly polarized and oriented perpendicular to **M** (see Supplementary Fig. 1).

Third, when reversing the layer sequence from F|N to N|F, $S_-(t)$ changes polarity (see Supplementary Fig. 2). Fourth, $S_-(t)$ does not depend on the pump-pulse polarization (linear and circular, see Supplementary Fig. 3). Finally, as seen in Fig. 2c, the root mean square (RMS) of $S_-(t)$ grows approximately linearly with the absorbed pump fluence. These observations are in line with the scenario suggested by Fig. 1.

**Impact of the metal layer**. To test the relevance of the ISHE, we replace the Pt with a tungsten (W) layer. The resulting $S_-(t)$ exhibits a reduced amplitude and reversed sign (Fig. 2d), consistent with previous ISHE works[26]. On bare YIG and single Pt

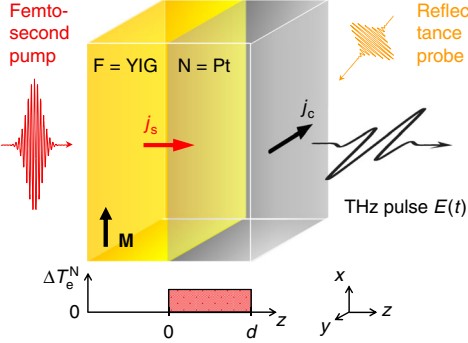

**Fig. 1** Experiment schematic. To probe the ultimate speed of the spin Seebeck effect, a femtosecond laser pulse (duration 10 fs, center photon energy 1.6 eV) is incident on a F|N bilayer made of $N$ = Pt (thickness of $d$ = 5 nm) on top of F = YIG (thickness 5 μm, in-plane magnetization **M**, electronic band gap of 2.6 eV). While the YIG film is transparent to the pump pulse, the Pt film is excited homogeneously, resulting in a transient increase $\Delta T_e^N$ of its electronic temperature. Any ultrafast spin–current density $j_s(t)$ arising in Pt is converted into a transverse charge–current density $j_c(t)$ by the inverse spin Hall effect, thereby acting as a source of a THz electromagnetic pulse whose transient electric field $E(t)$ is detected by electrooptic sampling. The electron dynamics in the Pt layer is interrogated by an optical probe pulse that measures the transient sample reflectance

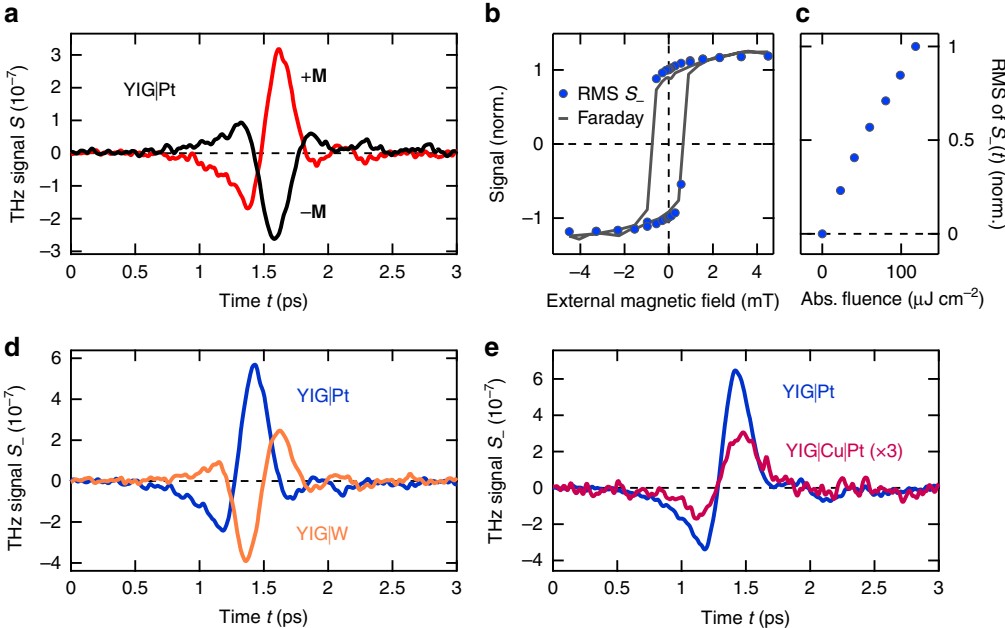

**Fig. 2** Terahertz emission of photoexcited F|N bilayers. **a** THz emission signals $S(\pm M)$ from a YIG(3 μm)|Pt(5.5 nm) sample for opposite directions of the in-plane YIG magnetization **M** as a function of time $t$. We focus on the difference $S_- = S(M) - S(-M)$ odd in **M**. **b** Amplitude of the THz signal $S_-$ (root-mean-square, RMS) and the Faraday rotation of a continuous-wave laser beam (wavelength 532 nm) as a function of the external magnetic field. Both hysteresis loops were measured under identical pump conditions at room temperature. **c** Amplitude of $S_-$ as a function of the absorbed pump fluence. **d** THz emission signal from a 3 μm thick YIG film capped with Pt and W, both 5.5 nm thick. **e** THz emission signal from a 5 μm YIG film capped with Pt(5.6 nm) or Cu(1.9 nm)|Pt(5.4 nm)

films, no signal $S_-(t)$ is detected above the noise floor (see Supplementary Fig. 4). These measurements provide supporting evidence that the femtosecond pump pulse injects an ultrafast, **M**-polarized spin current along the interface normal and into the N layer where the ISHE is operative (see Fig. 1).

Figure 2e shows that even when introducing a 1.9 nm copper (Cu) spacer layer between YIG and Pt, a measurable THz signal persists. Our result is fully consistent with the picture of a heat-induced spin current flowing from cold YIG into hot Pt, traversing the Cu layer. The presence of the Cu film decreases the current amplitude due to loss[18,27] and the reduced optical excitation density[18].

In summary, the THz emission signal $S_-$ exhibits all the characteristics expected for the SSE. We, therefore, regard the THz data of Fig. 2 as a manifestation of the SSE at THz frequencies. Our measurements rule out alternative THz emission scenarios: (i) An anomalous Nernst effect by proximity-induced moments in Pt would be quenched by a Cu spacer layer, in contrast to our data of Fig. 2e. This contribution is, therefore, negligibly small, in agreement with previous results[28,29]. (ii) Likewise, a photo-spin-voltaic effect[30] does not make a noticeable contribution to the THz signal. (iii) A THz signal due to the Nernst effect[2] in the N layer would be directly proportional to the external magnetic field, in contrast to our observations of Fig. 2b. (iv) Finally, optical orientation of spins by the optical pump beam in YIG[24] or Pt[31] would depend on the pump polarization, again in contrast to our observations.

**Temperature dependence**. As the SSE current depends on the ferrimagnet's magnetization **M** (see Fig. 2b), a marked temperature dependence of the THz emission signal is expected. Figure 3a displays the bulk magnetization of the YIG(3 μm)|Pt(5.5 nm) sample versus the ambient temperature $T_0$ as determined by the Faraday effect. The Faraday signal disappears at the Curie temperature $T_C = 550$ K of bulk YIG. Figure 3b reveals that the RMS

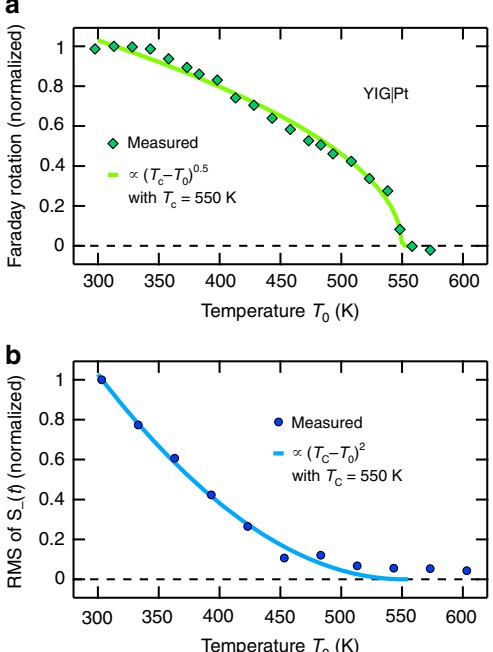

**Fig. 3** Effect of sample temperature. **a** Faraday rotation of a continuous-wave laser beam (wavelength 532 nm) after transmission through a YIG(3 μm)|Pt(5.5 nm) sample as a function of the sample temperature. A fit proportional to $(T_C - T_0)^\alpha$ (ref. [33]) yields a critical exponent of $\alpha = 0.5$ and a Curie temperature of $T_C = 550$ K. **b** Temperature dependence of the amplitude of the magnetic THz emission signal $S_-$, which differs from that of the sample magnetization, similar to previous work on the DC SSE[32]. The solid line is a fit proportional to $(T_C - T_0)^\alpha$, here yielding $\alpha = 2.0 \pm 0.5$

of $S\_(t)$ and, thus, the THz spin current also decreases with rising $T_0$, but more rapidly than the YIG bulk magnetization. A similar monotonic decrease was seen in static experiments on YIG|Pt bilayers, where a temperature gradient in the YIG bulk drives the spin current[32,33]. Fitting the model function $(T_C - T_0)^\alpha$ to our data yields an exponent of $\alpha = 2.0 \pm 0.5$ (Fig. 3b), close to the exponents 1.5 and 3 found in refs.[32,33], respectively. This agreement provides further evidence that the THz signal $S\_(t)$ arises from the ultrafast SSE.

**Ultrafast spin Seebeck current**. Figure 4a displays the THz signal $S\_(t)$ from a YIG(3 µm)|Pt(5.5 nm) bilayer, measured with a

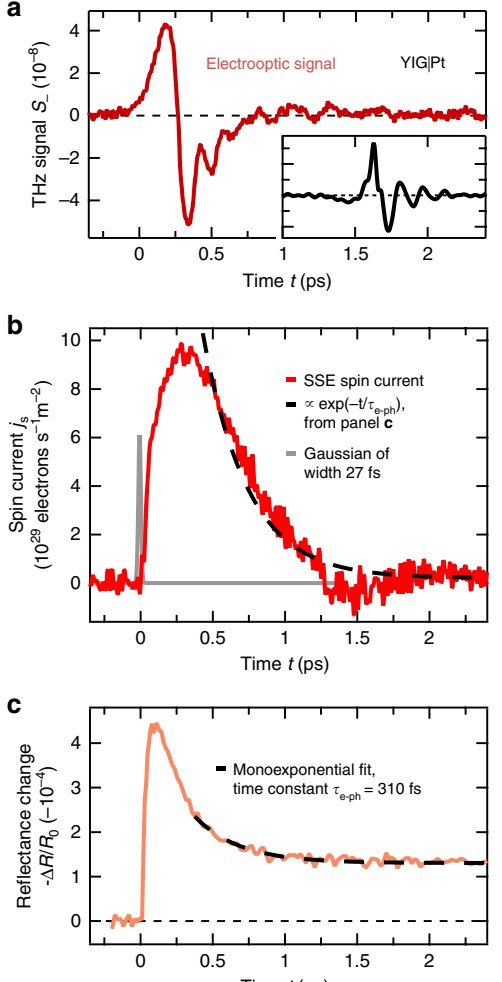

**Fig. 4** Ultimate speed of the spin Seebeck effect. **a** THz signal $S\_(t)$ from a YIG(3 µm)|Pt(5.5 nm) bilayer measured using a 250 µm thick GaP(110) electrooptic crystal. The inset shows the transfer function $h(t)$ relating the electrooptic signal $S\_(t)$ to the THz electric field directly behind the sample (Eq. (6)) and, thus, the spin current $j_s(t)$ within (Eq. (7)). Approximately, $h(t)$ acts like a temporal derivative on $j_s(t)$. **b** Extracted spin-current density $j_s(t)$ entering the Pt layer (red line). The gray line is a Gaussian with a full-width at half maximum of 27 fs and visualizes an upper limit to the experimental time resolution (see Methods). The dashed black line is the monoexponential decay as obtained from the pump-induced sample reflectance of panel (**c**). **c** Pump-induced relative changes $-\Delta R(t)/R_0$ in the reflectance of a Pt thin film under excitation conditions similar to those used for measuring the THz emission signal of panel (**a**) (orange line). The dashed line is a fit of a monoexponential decay plus an offset for $t > 350$ fs and yields a time constant of $\tau_{e-ph} = 310$ fs

broadband THz electrooptic crystal. It is related to the spin-current density $j_s(t)$ injected into the N layer by a convolution (Eq. (6)) with a transfer function $h(t)$. We determine $h(t)$ by a reference measurement (see Methods). Its shape (inset of Fig. 4a) implies that $S\_(t)$ is roughly proportional to the derivative of $j_s(t)$.

Knowledge of the transfer function allows us to apply an inversion procedure[25] to the THz signal waveform (Fig. 4a). We obtain the central experimental result of this study (Fig. 4b): the ultrafast dynamics of the SSE spin current $j_s(t)$ induced by an ultrashort laser pulse. The time resolution of the spin-current transient is better than 27 fs (see Fig. 4b, Supplementary Fig. 5 and Methods section). Note that $j_s(t)$ exhibits an ultrafast rise and decay on a time scale of ~100 fs, more than one order of magnitude faster than any SSE response time reported so far[8,9,22,23].

**Transient reflectance**. To identify the mechanisms underlying the ultrafast spin-current dynamics, we note that they are triggered by optical excitation of the Pt layer. It is, thus, instructive to briefly review the dynamic response of metal thin films to homogeneous ultrafast optical excitation[34]. Primarily, at $t = 0$, the absorbed pump energy is deposited in the electronic system, thereby inducing a nonequilibrium electron distribution. Due to electron–electron and electron–phonon scattering, the electrons approach a Fermi–Dirac distribution. Simultaneously, yet with a usually slower time constant $\tau_{e-ph}$, the hot electrons cool down by energy transfer to the phonon system.

A few hundreds of femtoseconds after optical excitation, electron and phonon subsystems can often be adequately described by temperatures. Their transient changes $\Delta T_e(t)$ and $\Delta T_{ph}(t)$ are monitored by considering the pump-induced change $\Delta R(t)$ in the sample reflectance, which scales approximately linearly with $\Delta T_e(t)$ and $\Delta T_{ph}(t)$ (ref.[35]). As seen in Fig. 4c, $-\Delta R(t)$ rises rapidly and relaxes toward a constant background. For $t > 350$ fs, the decay is well described by a monoexponential function with a time constant of $\tau_{e-ph} = 310 \pm 70$ fs (dashed line in Fig. 4c). We assign this relaxation to energy transfer from the electrons to the phonons. Remarkably, the spin current $j_s(t)$ exhibits a very similar decay (Fig. 4b). This observation strongly indicates that $j_s(t)$ quasi-instantaneously follows the transient changes $\Delta T_e(t)$ in the electron temperature on the time scale of electron cooling. It also suggests that the intrinsic response time of the SSE is significantly faster than $\tau_{e-ph}$.

**Dynamic SSE model**. To understand this surprisingly fast response and the nature of the initial rise of the spin current (Fig. 4b), we adapt the static SSE theory of ref.[3] to the dynamic case and employ a linear-response approach to spin pumping[36,37]. As detailed in the Methods section, our treatment is based on the microscopic model that is schematically shown in Fig. 5a. In the following, a concise and intuitive summary is given.

According to ab initio calculations[38], the spins of the interfacial F and N layers are coupled by an sd-exchange-like Hamiltonian[3,7,39,40] $J_{sd}\mathbf{S}^F \cdot \mathbf{S}^N$ over a thickness of about one YIG lattice constant $a = 1.24$ nm. Here, $J_{sd}$ quantifies the coupling strength, and $\hbar\mathbf{S}^F$ and $\hbar\mathbf{S}^N$ are the total electron spin angular momenta contained in an interfacial cell of dimension $a^3$ on the F and N side, respectively, with $\hbar$ denoting the reduced Planck constant. Thermal spin fluctuations $\mathbf{s}^F(t)$ in F and $\mathbf{s}^N(t)$ in N cause stochastic effective magnetic fields and, therefore, torques on each other, which cancel in thermal equilibrium.

However, this balance is broken in our experiment by the pump pulse exciting exclusively the N-cell electrons. Consequently, we focus on elementary interactions caused by spin fluctuations in N. After the arrival of the pump pulse, say at

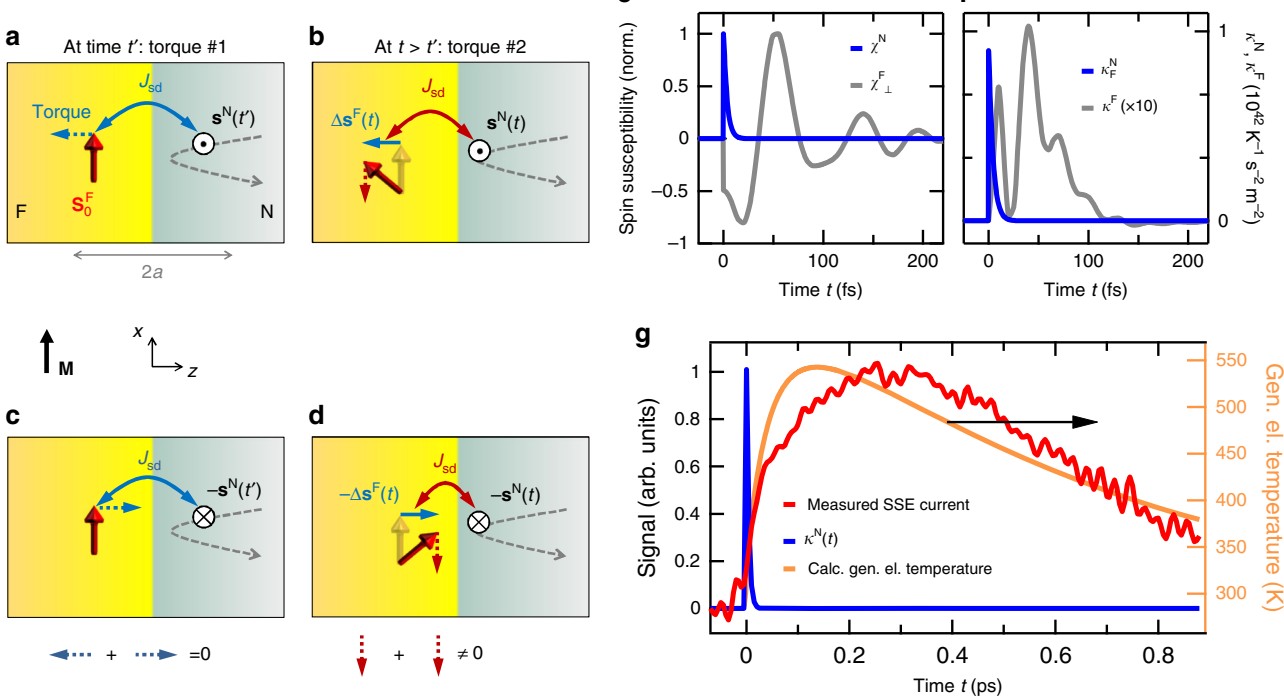

**Fig. 5** Dynamic SSE model. **a–d**, Model schematic of the F|N interface. To illustrate the action of the exchange torque exerted by N on F, it is sufficient to consider the "up" (⊙, see panels **a**, **b**) and "down" case (⊗, see panels **c**, **d**) of an N-cell spin fluctuation $s^N$ perpendicular to the YIG magnetization **M**. **a** At time $t'$, an N electron entering the interaction region induces a fluctuation $s^N(t')$ of the total N-cell spin, thereby exerting the effective magnetic field $J_{sd}s^N(t')$ on the adjacent F-cell spin (torque #1). **b** Consequently, at a slightly later time $t$, the F-cell spin has changed by $\Delta s^F(t)$ proportional to $J_{sd}s^N(t')$. **c** The opposite fluctuation $-s^N(t')$ at time $t'$ induces **d** the change $-\Delta s^F(t)$, resulting in zero change in the F-cell spin, $\langle \Delta s^F(t)\rangle = 0$. However, as seen in panels (**b**) and (**d**), a second interaction at $t > t'$ with the N-cell field (torque #2) leads to the same rectified torque $J_{sd}s^N(t) \times \Delta s^F(t)$ for both $+s^N$ and $-s^N$ and, thus, a net spin current between F and N. **e** Calculated time-domain spin susceptibility of the F cell (transverse $\chi^F_\perp(t)$ of YIG) and the N cell (isotropic $\chi^N(t)$ of Pt). **f** Calculated dynamics of the SSE response functions $\kappa^N(t)$ and $\kappa^F(t)$ which quantify, respectively, the spin current induced by a $\delta(t)$-like temperature change of the N (Pt) and F (YIG) layer. The area under both curves equals the DC SSE constant $\mathcal{K}$. **g** Evolution of the generalized electronic temperature of Pt as obtained by simulations based on the Boltzmann equation. Excitation conditions are similar to those used in the experiment. For direct comparison, the measured spin current $j_s(t)$ (see Fig. 4b) and calculated SSE response function $\kappa^N(t)$ (see panel **f**) are also shown

time $t'$, the conduction-electron spins within an N cell give rise to a random field $J_{sd}s^N(t')$ on the F-cell spins[3] (Fig. 5a). Subsequently, at time $t > t'$, the F-cell spin has changed dynamically by $\Delta s^F(t) = \underline{\chi}^F(t - t')J_{sd}s^N(t')$, where $\underline{\chi}^F = \left(\chi^F_{ij}\right)$ is the spin susceptibility matrix of the F cell[3,36] (Fig. 5b). However, $\Delta s^F(t)$ is canceled by an oppositely oriented field $-J_{sd}s^N(t')$ occurring with equal probability (see Fig. 5c, d). In other words, the average-induced moment $\langle \Delta s^F(t)\rangle$ vanishes because $\langle s^N\rangle = 0$.

Nevertheless, a net effect results from a second interaction of F with $J_{sd}s^N$ at time $t > t'$ (Fig. 5b, d). The corresponding torque $J_{sd}s^N(t) \times \Delta s^F(t)$ scales with $J^2_{sd}$ and, therefore, rectifies the random field $J_{sd}s^N$. Its expectation value is parallel to the F magnetization **M** (Fig. 5b, d). By integration over all first-interaction times $t'$ (see Methods section), we find the spin current due to N-cell fluctuations equals

$$j^N_s(t) = \frac{J^2_{sd}}{a^2}\int dt' \chi^F_\perp(t - t')\langle s^N_z(t)s^N_z(t')\rangle. \quad (2)$$

Equation (2) provides the key to understanding the ultrafast dynamics of the SSE. The spin-correlation function $\langle s^N_z(t)s^N_z(t')\rangle$ implies that a net spin current only arises if the two interactions with $J_{sd}s^N$ occur within the correlation time $\tau^N$ of the N-cell spin, that is, for $|t - t'| < \tau^N$. The $\tau^N$ can be estimated by the time it takes an electron to traverse the interaction region of width

$\sim a$ (Fig. 5a), yielding $\tau^N \sim 4\,\text{fs}$ for Pt. As this time constant is shorter than the pump-pulse duration, the N-cell spin correlation function mirrors the instantaneous state of the optically excited electrons in the metal.

Interestingly, the F-cell spins react instantaneously, too, because they have no inertia[41]. This fact is illustrated by the step-like onset of the transverse F-cell susceptibility $\chi^F_\perp(t) = \chi^F_{yz}(t) - \chi^F_{zy}(t)$ at $t = 0$ (Fig. 5e). Consequently, the spin current follows the dynamics of the electron distribution in the metal without delay.

**From fluctuations to generalized temperatures.** To put the last conclusion on a more quantitative basis, we note that the fluctuations of the N-cell spin derive from those of the flux of the N electrons incident on the F|N interface (see Fig. 5a–d and Methods). We, accordingly, expect that the strength of the current fluctuations scales with the number of available electronic scattering channels, that is, with the number of occupied initial and unoccupied final Bloch states. Indeed, the variance of the current noise is known to be proportional to[42]

$$\int d\epsilon\, n(\epsilon, t)[1 - n(\epsilon, t)]D(\epsilon) = k_B \tilde{T}^N_e(t)D(\epsilon_F), \quad (3)$$

where $n(\epsilon, t)$ is the occupation number of an electron Bloch state at energy $\epsilon$, $D(\epsilon)$ denotes the density of Bloch states, $k_B$ is the Boltzmann constant, and $\epsilon_F$ is the Fermi energy.

When $n(\epsilon, t)$ is a Fermi–Dirac function at temperature $T_e^N(t)$, the term $n(\epsilon, t)[1 - n(\epsilon, t)]$ peaks at $\epsilon = \epsilon_F$ with width $4k_B T_e^N(t)$ and height $1/4$. Therefore, the quantity $\tilde{T}_e^N(t)$ introduced by Eq. (3) becomes equal to $T_e^N(t)$, provided $D(\epsilon)$ is constant around $\epsilon_F$. The remarkable equality $\tilde{T}_e^N(t) = T_e^N(t)$ is still satisfied to a good approximation even for the strongly energy-dependent density of states of Pt at all electronic temperatures relevant to our experiment (see Supplementary Fig. 6). We, consequently, identify $\tilde{T}_e^N(t)$ as a generalized electronic temperature that is applicable to arbitrary nonthermal electron distributions.

Using linear-response theory (see Methods), we can, thus, express the correlation function $\langle s_z^N(t)s_z^N(t')\rangle$ by means of $\tilde{T}_e^N$ and the isotropic spin susceptibility $\chi^N$ of the N cell. As the F layer remains cold at temperature $T_0$, we obtain

$$j_s(t) = j_s^N(t) = \int dt'\,\kappa^N(t - t')\Delta\tilde{T}_e^N(t'), \qquad (4)$$

where $\Delta\tilde{T}_e^N = \tilde{T}_e^N - T_0$ is the pump-induced increase of the electron temperature of N. Note that Eq. (4) is the desired generalization of Eq. (1) for time-dependent temperatures and nonthermal electron distributions of the N layer.

The response function $\kappa^N \propto J_{sd}^2\chi^N(t)\int dt'\chi_\perp^F(t - t')\chi^N(t')$ can be understood as the spin current induced by a $\delta(t)$-like change in $\tilde{T}_e^N$. It is determined by the susceptibilities of the F-cell and N-cell spins. For N = Pt, we assume an isotropic spin susceptibility[43,44] $\chi^N(t)$ that rises step-like and decays with time constant $\tau^N$ (see Eq. (26) and Fig. 5e). In contrast, $\chi_\perp^F(t)$ is obtained by atomistic spin-dynamics simulations[12,45] and exhibits a strongly damped oscillation reflecting the superposition of numerous magnon modes (see Methods and Fig. 5e). The resulting SSE response function $\kappa^N(t)$ is shown in Fig. 5f.

**Comparison to measured and simulated electron dynamics.** As expected from our qualitative discussion following Eq. (2), $\kappa^N(t)$ (Fig. 5f) has an ultrashort duration on the order of $\tau^N$, much faster than the onset of the measured spin current (Fig. 5g). Therefore and because of Eq. (4), the spin current quasi-instantaneously follows the dynamics of the generalized electron temperature of N,

$$j_s(t) = \mathcal{K}\Delta\tilde{T}_e^N(t), \qquad (5)$$

where $\mathcal{K} = \int dt\,\kappa^N(t) = \int dt\,\kappa^F(t)$ is the static SSE coefficient. We now put this conclusion to test by considering the rise and decay dynamics of the measured spin current $j_s(t)$ (Fig. 4b).

First, Eq. (5) is fully consistent with the decay of $j_s(t)$ for $t > 350$ fs (Fig. 4b), whose evolution agrees well with the cooling dynamics of the electron bath due to electron–phonon coupling (Fig. 4c). This agreement shows that Eqs. (4) and (5) are valid for thermal electron distributions and on time scales significantly faster than $\tau_{e-ph} = 310 \pm 70$ fs.

Second, to check Eq. (5) over the phase in which the measured $j_s(t)$ rises, we simulate the electron population dynamics $n(\epsilon, t)$ using the Boltzmann equation for excitation conditions close to those in our experiment. Optical excitation, electron–electron and electron–phonon scattering are explicitly taken into account by collision integrals[46,47] (see Methods). The evolution of the generalized electron temperature $\Delta\tilde{T}_e^N(t)$ is calculated through Eq. (3) and shown in Fig. 5g. Note that the increase of $\Delta\tilde{T}_e^N(t)$ proceeds within ~100 fs, which is much slower than the duration of the pump pulse and the width of the SSE response function $\kappa^N(t)$. The evolution of $\Delta\tilde{T}_e^N(t)$ agrees well with that of the measured $j_s(t)$ (Fig. 5g), thereby confirming the validity of Eqs. (4) and (5) for time scales much shorter than 100 fs and for nonthermal electron distributions.

To understand the noninstantaneous rise of the generalized electron temperature, we note that $\Delta\tilde{T}_e^N(t)$ approximately scales with $\int_{\epsilon > \epsilon_F} d\epsilon\,\Delta n(\epsilon, t)$, that is, the pump-induced number of electrons above the Fermi energy $\epsilon_F$ (Eq. (3)). Initially, photoexcitation induces electrons and holes at approximately half the pump photon energy of $\hbar\omega_p = 1.6$ eV away from the Fermi energy. However, subsequent scattering cascades lead to thermalization of the electrons, thereby generating roughly $\hbar\omega_p/k_B T_0 \sim 60$ thermal electron–hole pairs out of each initially photoinduced pair. This carrier multiplication, in turn, strongly increases the generalized temperature and the spin current (Fig. 5g). It can be considered as an experimental confirmation of the notion that the SSE current is due to electrons impinging on the YIG|Pt interface[37–50].

## Discussion

The ~100 fs time scale of electron thermalization in Pt as observed here is consistent with time-resolved photoelectron spectroscopy of Pt at roughly comparable excitation densities[51]. Similarly, for Ru, another transition metal, the number of photoinduced electrons above the Fermi energy[52] and, thus, $\Delta\tilde{T}_e^N(t)$ was observed to rise on a time scale of 100 fs for quite similar excitation densities (Fig. 8 in ref. [52]). We note that for a more free-electron-like metal, such as Al, in contrast, electron thermalization is known to proceed significantly faster because of the smaller Coulomb screening parameter[47].

While the preceding analysis has focused on the time scales of the SSE current, we now consider the magnitude of the measured and simulated spin current. In our experiment, the SSE efficiency is given by the THz peak field divided by the peak increase of the generalized electron temperature (Fig. 5g) and estimated to be ~2 V m$^{-1}$ K$^{-1}$. This value is comparable to results from SSE experiments on samples with Pt layers of similar thickness, that is, for static heating (0.1 V m$^{-1}$ K$^{-1}$)[53] and laser heating at MHz (0.7 V m$^{-1}$ K$^{-1}$)[8] or GHz frequencies (37 V m$^{-1}$ K$^{-1}$)[9]. Our modeling also allows us to extract the YIG|Pt interfacial exchange coupling constant, yielding $J_{sd} \sim 2$ meV or Re $g^{\uparrow\downarrow} \sim 1 \times 10^{18}$ m$^{-2}$ in terms of the spin-mixing conductance $g^{\uparrow\downarrow}$, in good agreement with calculated[38] and measured values[54,55].

We note that the positive sign of the measured spin current $j_s$ (Fig. 4b) implies that the magnetization of YIG decreases. The integrated $j_s(t)$ is equivalent to increasing the temperature of the thin YIG interfacial layer of thickness $a$ by at most ~50 K (see Fig. 3a). As this value is significantly smaller than the increase of the Pt electron temperature, we can neglect the back-action of the heated YIG layer on the spin current.

Equations (7) and (28) of our analytical theory (see Methods) allow us to discuss the dependence of the THz SSE amplitude on temperature $T_0$ (Fig. 3b) in more detail. If we assume that the spin-current relaxation length in Pt scales linearly with the Pt conductivity, neglect the small variations of the THz impedance $Z(\omega)$ versus $T_0$ and note that the spin Hall conductivity of Pt is approximately constant over the temperature range considered here[56,57], Eq. (7) implies that the $T_0$ dependence of the THz SSE signal originates exclusively from the spin-Seebeck current $j_s$. Since the N-layer spin susceptibility is not expected to vary with temperature[43], Eq. (28), in turn, shows that the temperature dependence of $j_s$ is governed by that of the product $J_{sd}^2\chi_\perp^F(t = 0^+)$ of the interface exchange-coupling constant and F-cell spin susceptibility. Indeed, previous work[58–60] has provided strong indications that the temperature dependence of the spin susceptibility at interfaces can differ strongly from that of the bulk magnetization and that the interlayer exchange-coupling parameter may be influenced by the temperature of the spacer layer.

So far, our experiments have been restricted to excitation of the metal part of YIG|Pt. Our modeling, however, allows us to also calculate the SSE response function $\kappa^F(t)$ related to heating of the F = YIG layer (see Eq. (18)). Note that $\kappa^F(t)$ exhibits clear features of the susceptibility of the F-cell spins (Fig. 5f). Measurement of $\kappa^F(t)$ would, therefore, provide insights into magnon dynamics on the unit-cell level. If YIG and Pt layers were uniformly and simultaneously heated by a sudden temperature jump, static SSE theory (Eq. (1)) would imply a vanishing current. In contrast, our theory predicts a 100 fs short current burst (Supplementary Fig. 7), which reflects the inherent asymmetry of the F|N structure. At times $t > 100$ fs, the total spin current vanishes, consistent with the familiar static result of Eq. (1).

In conclusion, we measured an ultrafast spin current in the prototypical SSE system YIG|Pt triggered by femtosecond optical excitation of the metal layer. The current exhibits all the hallmarks expected from the THz SSE. Our dynamic model, based on sd-like exchange-coupled YIG|Pt layers, can reproduce both the magnitude and the dynamics of the measured ultrafast spin current. It allows us to identify the ultrafast elementary steps leading to the formation of the initial SSE current: optically excited metal electrons impinge on the interface with the magnetic insulator. They apply random torque that is rectified by two subsequent interactions, thereby resulting in a net spin current from YIG into the metal.

The SSE response to heating of the metal layer is quasi-instantaneous for two reasons. First, the total electron spin of a Pt unit cell at the YIG|Pt interface has a correlation time of less than 4 fs. Second, the YIG spins respond to these fluctuations without inertia. We emphasize that the step-like impulse response of the YIG spins is a feature of all ferromagnetic magnons of YIG and independent of their frequencies, be it megahertz or THz. As a consequence of these instantaneous responses, the SSE current directly monitors the thermalization and cooling of the photoexcited electrons, which both proceed on a sub-picosecond time scale.

In terms of applications, the observed ultrafast SSE current can be understood as a first demonstration of incoherent THz spin pumping. Therefore, an instantaneously heated metal layer is a promising transducer for launching ultrashort incoherent THz magnon pulses into magnetic insulators. They may prove useful for magnon-based transport of information, for exerting ultrafast torques on remote magnetic layers[61] and for spectroscopy of spin waves with nanometer wavelength[62]. Our results also strongly suggest that coherent spin pumping should be feasible at THz frequencies[63].

From a fundamental viewpoint, our experimental approach permits the characterization of the interfacial SSE and the ISHE of metals in standard bilayer thin-film stacks with a large sample throughput and without extensive micro-structuring[64]. It allows one to all-optically probe the magnetic texture and the exchange coupling at interfaces. Since our setup is driven by a femtosecond laser oscillator rather than a significantly more demanding amplified laser system, our methodology should be accessible to a broad community. As indicated by Eq. (2), the THz SSE current is also sensitive to the local electron-spin noise at the highest frequencies, even under conditions far from equilibrium. Such type of spin-noise spectroscopy is difficult to realize with other methods[42].

We finally emphasize that the SSE is a model case of incoherent angular-momentum transfer between a spin ensemble and another system[65], such as the electronic orbital degrees of freedom, the crystal lattice or a second spin sublattice of a solid. Therefore, our modeling may serve as a blueprint for a large variety of optically driven spin dynamics, for instance ultrafast switching[66] and quenching[67] of magnetic order. Our insights highlight the significant role of carrier multiplication in these processes and strongly suggest that lower pump photon energies (ideally on the order of the thermal energy) will substantially shorten the rise time of the angular-momentum transfer and extend its bandwidth to tens of THz.

## Methods

**Sample preparation.** The YIG films (thicknesses of 2, 3 and 5 μm) were grown by liquid-phase epitaxy on 500 μm thick gadolinium gallium garnet (GGG) substrates (Innovent e.V., Jena, Germany). The YIG surface was cleaned with isopropyl alcohol and acetone in an ultrasonic bath. In situ argon etching was omitted in order to maintain the integrity of the YIG surface magnetization[68].

Subsequently, films of Pt, W and MgO were grown on YIG using the Singulus Rotaris sputter deposition system. The MgO serves as a protection against oxidation for the W film. Pt and W were grown using DC magnetron sputtering whereas radio-frequency sputtering was used for MgO growth from a composite target. The deposition rates for Pt, W and MgO were 3.1, 1.5 and 0.08 Å s$^{-1}$, respectively, at a pressure of $5.7 \times 10^{-3}$, $3.5 \times 10^{-3}$ and $1.8 \times 10^{-3}$ mbar. For the measurements displayed in Fig. 2e, Pt and Cu layers were grown using a home-built deposition system with DC magnetron sputtering at rates of 0.7 and 0.63 Å s$^{-1}$, respectively, at a pressure of 0.01 and 0.025 mbar.

The samples were characterized magnetooptically by the Faraday effect of a beam from a 512 nm laser diode under an angle of incidence of 45°. In this way, hysteresis loops were measured by slowly varying the external magnetic field.

**THz-emission setup.** In the optical experiment, the in-plane sample magnetization was saturated by an external magnetic field of 10 mT. For setting the sample temperature $T_0$ between 300 and 600 K, a resistive heating coil was attached to the sample holder onto which the sample was glued with a heat-conducting silver paste. The temperature was measured with a type-K thermocouple.

As schematically shown in Fig. 1, the sample was excited by linearly or circularly polarized laser pulses (duration 10 fs, center wavelength 800 nm, pulse energy 2.5 nJ) from a Ti:sapphire laser oscillator (repetition rate 80 MHz) under normal incidence from the GGG/YIG side (beam diameter at sample 22 μm full-width at half-maximum of the intensity). The resulting absorbed fluence was 120 μJ cm$^{-2}$.

The duration of the pump pulse arriving in the Pt layer was optimized by adjusting the optical thickness of a pair of wedged prisms and the number of reflections on a pair of chirped mirrors. As signal, we used the photocurrent generated by the pulse train in a 2-photon-absorption photodiode behind a 0.5 mm thick BK7 substrate (group-delay dispersion of 22 fs$^{-2}$). In the experiment, the BK7 window was replaced by a 0.5 mm thick GGG substrate[69] with a group-delay dispersion of 82 fs$^{-2}$, yielding an upper bound of 19 fs for the pulse duration.

The THz electric field emitted in transmission direction was detected by electrooptic sampling[70,71], where probe pulses (0.6 nJ, 10 fs) from the same laser copropagate with the THz field through an electrooptic crystal. The resulting signal $S(t)$ equals twice the THz-field-induced probe ellipticity, where $t$ is the delay between the THz and probe pulse. Depending on the signal strength and bandwidth required, we used various electrooptic materials, ZnTe(110) (thickness of 1 mm) and GaP(110) (250 μm). If not mentioned otherwise, all measurements were performed at room temperature in a dry N$_2$ atmosphere.

**Extraction of the THz current.** Generally, the measured electrooptic signal $S(t)$ is related to the THz electric field $E(t)$ directly behind the sample by the convolution

$$S(t) = (h * E)(t) = \int dt' h(t - t') E(t'). \tag{6}$$

Here, the transfer function $h(t)$ accounts for propagation to the detection unit, as well as the detector response function of the electrooptic-sampling process. We determined this function by using an appropriate reference emitter[25].

The measured transfer function (inset of Fig. 4a) exhibits a sharp bipolar feature around $t = 0$, which upon convolution with $E(t)$ approximately yields a signal proportional to the derivative of the field, $S(t) \propto \partial E(t)/\partial t$. Equation (6) was inverted directly in the time domain by recasting it as a matrix equation. Note that the DC component of $h(t)$ is zero because a DC electric field cannot propagate away from its source. We determined the missing DC component of $E(t)$ by using the causality principle: the pump-induced charge current inside the sample and, thus, $E(t)$ is zero before arrival of the pump pulse at $t = 0$.

In the frequency domain, the field $E(\omega)$ is related to the spin current injected into the Pt layer by a generalized Ohm's law[17]

$$E(\omega) = eZ(\omega)\theta_{SH}\lambda_{rel} j_s(\omega). \tag{7}$$

Here, $\omega/2\pi$ denotes frequency, and $-e$ is the electron charge. The impedance $Z(\omega)$ of the YIG|Pt bilayer on GGG was determined by THz transmission spectroscopy[17]. The spin Hall angle of Pt is assumed to be $\theta_{SH} = 0.1$ (ref.[72]), and the relaxation length of the ultrafast spin current is $\lambda_{rel} = 1$ nm (ref.[17]).

Consequently, all transfer functions relating $S$, $E$ and eventually $j_s$ are known, and we can extract $j_s(t)$ from the measured THz signal $S(t)$ by inverting Eqs. (6) and (7)[25]. The polarization of the spin current was calibrated by using a metallic reference emitter[17].

Our inversion procedure is illustrated in Fig. 4a, b and Supplementary Fig. 5. For example, the electrooptic signal $S_-(t)$ odd in the YIG magnetization (Fig. 4a) approximately scales with the derivative of the field $E(t)$ and, thus, the charge-current density $-e\theta_{SH}j_s(t)$ (Fig. 4b). We also analyze the THz signal $S_+(t)$ even in the YIG magnetization (Supplementary Fig. 5c) and find a charge current that equals a sharp Gaussian peak with a width of 27 fs (Supplementary Fig. 5d). We infer that the time resolution of the spin-current transient is better than 27 fs. This value is a result of the pump-pulse duration in the Pt layer and of the low-pass filtering included in our extraction procedure, which imply a temporal broadening of at most 19 and 24 fs, respectively.

**Transient reflectance**. To conduct optical-pump reflectance-probe measurements on a Pt thin film (thickness of 30 nm), the beam of p-polarized laser pulses (duration 14 fs, center wavelength 800 nm) from a cavity-dumped Ti:sapphire oscillator (repetition rate 1 MHz) was split into pump and probe pulses at a power ratio of 4. The pump and probe beams were incident onto the sample at angles of 45° and 50°, respectively. The pump-induced modulation of the reflected probe power was measured using a photodiode and lock-in detection and yielded the relative pump-induced change $\Delta R(t)/R_0$ in the sample reflectance. The pump-pulse parameters, the thickness of the Pt film (30 nm), and the absorbed pump fluence (400 μJ cm$^{-2}$) were chosen such to obtain excitation conditions similar to those used for measuring the spin current (Fig. 4b).

If electrons and phonons of the photoexcited metal film can be adequately described by temperatures and their transient changes $\Delta T_e(t)$ and $\Delta T_{ph}(t)$ are small, the pump-induced change $\Delta R(t)$ in the sample reflectance scales approximately linearly with $\Delta T_e(t)$ and $\Delta T_{ph}(t)$ (ref.[35]). In the absence of transport, energy conservation furthermore implies $\Delta \dot{T}_e(t) \propto \dot{T}_{ph}(t)$, and $\Delta R(t)$ becomes proportional to $\Delta T_e(t)$ plus a constant.

**Temperature estimates**. The peak electronic temperature induced by the pump pulse is obtained from the simulations of the dynamics of the electron distribution function (see below) and the resulting evolution of the generalized temperature (see Eq. (3) and Fig. 5g). To estimate how strongly the YIG is modified by the ultrafast spin current, we time-integrate the measured $j_s(t)$ (Fig. 4b). The resulting loss of spin angular momentum reduces the magnetization of the first YIG monolayer by 5%, which is equivalent to increasing its temperature by about 50 K (see Fig. 3a). Material parameters relevant for these estimates are summarized in Supplementary Table 1.

**Derivation of the SSE current**. As illustrated in Fig. 5a–d, interfacial F and N layers of thickness $a$ are coupled by nearest-neighbor sd-type exchange interaction. We divide the interfacial plane in N cells of size $a^3$ and consider the total electron spin $\hbar S_\alpha^F$ and $\hbar S_\alpha^N$ contained in an F cell and N cell with index $\alpha$, respectively. The expectation value of $S_\alpha^F$ is related to the F magnetization by $\langle S_\alpha^F \rangle \propto a^3 M$.

According to ab initio simulations[38], coupling between F and N spins is given by the sd-exchange-type Hamiltonian[3,7,39,40] $H_{sd} = J_{sd} \sum_\alpha S_\alpha^F \cdot S_\alpha^N$. Therefore, each $S_\alpha^N$ applies the torque $S_\alpha^F(t) \times J_{sd}S_\alpha^N(t)$ on the adjacent $S_\alpha^F$. Accordingly, the total $H_{sd}$-related torque exerted by N on F is given by the sum over all cells $\alpha$. By taking the expectation value, we obtain the average spin-current density

$$\mathbf{j}_s = -\frac{J_{sd}}{Na^2} \sum_\alpha \langle S_\alpha^F \times S_\alpha^N \rangle \quad (8)$$

flowing from F to N, where $Na^2$ is the coupled interface area. Note that the tensor of the spin-current density is given by the tensor product $\mathbf{j}_s \otimes \mathbf{n} = \mathbf{j}_s^t \mathbf{n}$ with $\mathbf{n}$ being the normal unit vector of the F|N interface.

We now split the random observable $S_\alpha^F = \langle S_\alpha^F \rangle + s_\alpha^F + \Delta s_\alpha^F$ in three contributions: its mean value $\langle S_\alpha^F \rangle \propto a^3 M$ and its fluctuating part $s_\alpha^F$, both taken in the absence of interfacial coupling. In contrast, $\Delta s_\alpha^F$ quantifies the modification due to sd-coupling to the N layer. By applying an analogous splitting to $S_\alpha^N$, the spin current becomes

$$\mathbf{j}_s = \mathbf{j}_s^N + \mathbf{j}_s^F = \frac{J_{sd}}{Na^2} \sum_\alpha \langle s_\alpha^N \times \Delta s_\alpha^F - s_\alpha^F \times \Delta s_\alpha^N \rangle. \quad (9)$$

It has contributions $\mathbf{j}_s^N$ and $\mathbf{j}_s^F$ arising from spin fluctuations in N and F, respectively, which cancel in equilibrium. We approximate $\Delta s_\alpha^F$ to first order in $J_{sd}$, that is, by the linear response given by the spatiotemporal convolution[36]

$$\Delta s_\alpha^F = J_{sd} \sum_{\alpha'} \int dt' \underline{\chi}^F(\mathbf{x}_\alpha - \mathbf{x}_{\alpha'}, t - t') s_{\alpha'}^N(t'). \quad (10)$$

Here, $\underline{\chi}^F(\mathbf{x}, t) = (\chi_{ii'}^F(\mathbf{x}, t))$ is the spin susceptibility tensor of the F cell in matrix notation. An analogous expression holds for $\Delta s_\alpha^N$ with respect to $\chi^N(\mathbf{x}_\alpha - \mathbf{x}_{\alpha'}, t - t')s_{\alpha'}^N(t')$. We furthermore assume that spins of different cells have negligible

correlation, for instance $\langle s_{\alpha,i}^N(t)s_{\alpha',i'}^N(t') \rangle \propto \delta_{\alpha\alpha'}$. By substituting $\Delta s_\alpha^N(t)$ and $\Delta s_\alpha^F(t)$ in Eq. (9), we obtain

$$\mathbf{j}_s(t) = \mathbf{j}_s^N + \mathbf{j}_s^F = \frac{J_{sd}^2}{a^2} \int dt' \left[ \langle s^N(t) \times \underline{\chi}^F(t-t')s^N(t') \rangle - \langle s^F(t) \times \underline{\chi}^N(t-t')s^F(t') \rangle \right], \quad (11)$$

where $s^N = s_\alpha^N$ and $s^F = s_\alpha^F$ are the spins of any conjoined F and N cells, say $\alpha = \alpha' = 1$. The $\underline{\chi}^F(t) = \underline{\chi}^F(\mathbf{x}_\alpha - \mathbf{x}_\alpha = 0, t)$ and $\underline{\chi}^N(t) = \underline{\chi}^N(\mathbf{x}_\alpha - \mathbf{x}_\alpha = 0, t)$ can be interpreted as the spin susceptibility of any F and N cell, respectively. Therefore, we have arrived at the picture of a single pair of coupled F–N cells as considered in the main text (Fig. 5a).

In Eq. (11), the difference of the two terms reflects the competition between the torques arising from the fluctuations of the N-cell and F-cell spins. For example, as illustrated by Fig. 5a, b, the first term can be understood as follows: the fluctuating exchange field $J_{sd}s^N(t)$ due to N exerts torque on the magnetic moment $\Delta s^F(t) = J_{sd} \int dt' \underline{\chi}^F(t-t')s^N(t')$, which it has induced in F before. As this torque scales quadratically with the noise $s^N$, it does not vanish, provided $\Delta s^F(t)$ results from an earlier time $t'$ that lies inside the correlation window of the N-cell spin.

The last statement becomes more apparent when we explicitly write out the matrix and vector products in Eq. (11). Consequently, the component $j_s = j_{sx}$ of the spin-current density polarized along the sample magnetization $M$ (see Fig. 1) is found to be

$$j_s(t) = \frac{J_{sd}^2}{a^2} \sum_{jkl} \epsilon_{xjk} \int dt' \left[ \chi_{kl}^F(t-t') \langle s_j^N(t)s_l^N(t') \rangle - \chi_{kl}^N(t-t') \langle s_j^F(t)s_l^F(t') \rangle \right] \quad (12)$$

where $\epsilon_{xjk}$ denotes the Levi–Civita symbol. The first term of Eq. (12) quantifies the torque due to N-cell spin fluctuations and depends critically on the spin correlation function $\langle s_j^N(t)s_l^N(t') \rangle$ which typically peaks sharply around time $t = t'$. Any temperature change of the N spins will lead to a (possibly delayed) modification of the spin correlation function and, therefore, a spin-current response whose time dependence is determined by the spin susceptibility $\chi_{kl}^F(t)$ of the ferromagnet F. An analogous interpretation applies to the second term.

Equation (12) couples the dynamics proceeding in F and N. To describe dynamics in the bulk of F and N, Landau–Lifshitz–Bloch or spin-diffusion-type equations, respectively, can be used[3,73]. Note that Eq. (12) is quite generally valid, including the cases of insulating antiferromagnetic F layers and nonthermal states of F and N. If N is isotropic (as is fulfilled for Pt), one has $\langle s_j^N(t)s_l^N(t') \rangle \propto \delta_{jl} \langle s_z^N(t)s_z^N(t') \rangle$, and the first term of Eq. (12) turns into Eq. (2) of the main text. In the following, we first consider thermal states and subsequently extend our treatment to nonthermal electron distributions in the N layer.

**From fluctuations to temperatures**. To relate the correlation functions in Eq. (12) to temperatures in F and N, we consider the Kubo form of the fluctuation-dissipation theorem in the classical limit[74],

$$\langle s_i^N(t)s_j^N(t') \rangle = k_B T^N \cdot \left( \bar{\Theta} * \chi_{ij}^N - \Theta * \bar{\chi}_{ji}^N \right)(t-t'), \quad (13)$$

where $\Theta$ is the Heaviside step function. The overbar denotes time inversion, that is, $\bar{f}(t) = f(-t)$, and * denotes convolution (see Eq. (6)). Note that strictly this equation refers to equilibrium and cannot be applied to the situation of our experiment where the temperature of N (and F) is generally time-dependent.

To derive a fluctuation-dissipation theorem for a nonstationary system, we make use of the Langevin theory[45,74–76], in which the N-cell spin is assumed to be coupled to a bath of time-dependent temperature $T^N(t)$. In this framework[74,75], the spin fluctuations $s^N(t)$ arise from a random magnetic field $r^N(t)$ the bath applies to the spin system. Assuming $r^N$ has no memory and vanishing ensemble average, the intensity of the spin fluctuations is directly proportional to the instantaneous bath temperature,

$$\langle r_i^N(t)r_j^N(t') \rangle = A^N k_B T^N(t)\delta_{ij}\delta(t-t'), \quad (14)$$

where the constant $A^N$ quantifies how strongly the N bath and the N spins are coupled. By using linear response,

$$s^N(t) = \left( \underline{\chi}^N * r^N \right)(t), \quad (15)$$

and writing out the convolution (Eq. (6)), we obtain the spin–spin correlation function for a time-dependent bath temperature $T^N$,

$$\langle s_i^N(t)s_j^N(t') \rangle = A^N k_B \sum_m \int d\tau \chi_{im}^N(t-\tau)\chi_{jm}^N(t'-\tau)T^N(\tau). \quad (16)$$

This relationship shows that the temporal structure of the spin susceptibility $\chi_{ij}^N$ of N determines how quickly the system adapts to a sudden change in $T^N$. In the case of time-independent $T^N$, Eq. (16) reduces to the familiar Langevin-version

of the fluctuation–dissipation theorem[74,75]. Comparison with the Kubo-type fluctuation–dissipation theorem (Eq. (13)) yields

$$\bar{\Theta} * \left(\chi_{jl}^{N} - \overline{\chi}_{lj}^{N}\right) = A^{N} \sum_{m} \chi_{jm}^{N} * \overline{\chi}_{lm}^{N}. \tag{17}$$

This constraint on the spin susceptibility function can be used to determine the constant $A^{N}$. Completely analogous equations are obtained for the F-cell spin. We now substitute Eq. (16) and its analog for F into Eq. (12) and obtain

$$j_{s}(t) = \left(\kappa^{F} * T^{F} - \kappa^{N} * T^{N}\right)(t) \tag{18}$$

with the response functions

$$\kappa^{N}(t) = \frac{k_{B}J_{sd}^{2}A^{N}}{a^{2}} \sum_{jklm} \epsilon_{zjk} \chi_{jm}^{N}(t) \cdot \left(\chi_{kl}^{F} * \chi_{lm}^{N}\right)(t) \tag{19}$$

and, completely analogously,

$$\kappa^{F}(t) = \frac{k_{B}J_{sd}^{2}A^{F}}{a^{2}} \sum_{jklm} \epsilon_{zjk} \chi_{jm}^{F}(t) \cdot \left(\chi_{kl}^{N} * \chi_{lm}^{F}\right)(t). \tag{20}$$

Equations (18–20) can be considered as time-dependent generalization of the static constitutive relation (Eq. (1)) of the interfacial SSE. It can be shown that in the static limit of time-independent temperatures, Eq. (18) reduces to Eq. (1), that is, $j_{s} = \mathcal{K} \cdot (T^{F} - T^{N})$, where $\mathcal{K} = \int dt \kappa^{F}(t) = \int dt \kappa^{N}(t)$. The proof makes use of Eqs. (19) and (20), Parseval's theorem, Eq. (17) and the causality of the spin susceptibilities of F and N.

For an isotropic nonmagnetic metal N with $\chi_{ij}^{N}(t) = \delta_{ij}\chi^{N}(t)$, Eq. (19) implies the somewhat simpler relationship

$$\kappa^{N} = \frac{k_{B}J_{sd}^{2}A^{N}}{a^{2}} \chi^{N} \cdot \left[\left(\chi_{yz}^{F} - \chi_{zy}^{F}\right) * \chi^{N}\right]. \tag{21}$$

For the second response function, we obtain

$$\kappa^{F} = \frac{k_{B}J_{sd}^{2}A^{F}}{a^{2}} \sum_{m} \left[\chi_{ym}^{F} \cdot \left(\chi^{N} * \chi_{xm}^{F}\right) - \chi_{xm}^{F} \cdot \left(\chi^{N} * \chi_{ym}^{F}\right)\right]. \tag{22}$$

As seen from Eq. (21), the longitudinal spin susceptibility $\chi_{jj}^{F}$ of the F cell does not contribute to $\kappa^{N}$. The reason is that spin fluctuations along different coordinate axes are uncorrelated in the isotropic N layer. For example, the first interaction of the F layer with $J_{sd}s_{j}^{N}(t')$ would induce a change $\propto \chi_{jj}^{F}(t-t')J_{sd}s_{j}^{N}(t')$ in the F-cell spin, which is parallel to the $j$ axis. Because $\langle s_{j}^{N}(t)s_{l}^{N}(t')\rangle \propto \delta_{jl}$, the only relevant second interaction is due to $J_{sd}s_{j}^{N}(t)$, again along the $j$ axis. Therefore, no torque results, and the longitudinal $\chi_{jj}^{F}$ does not contribute to $\kappa^{N}$. This cancellation does not occur for $\kappa^{F}$ because spin fluctuations in F are correlated in different $j$ directions.

**Nonthermal electron distributions.** Our previous considerations, in particular Eq. (14), presume a thermal bath with temperature $T^{N}$. To reveal the nature of the bath and to also account for the nonthermal state of the N electrons directly after laser excitation, we extend our model of the N layer. As the fluctuation of the N-cell spin is assumed to arise predominantly from electrons entering and leaving the N cell[37,48–50], we model the dynamics of the N-cell spin as

$$s_{z}^{N}(t) \propto \left[\left(i_{\uparrow}^{in} - i_{\downarrow}^{in}\right) * p\right](t). \tag{23}$$

Here, $i_{\sigma}^{in}(t)$ is the current of electrons with spin $\sigma = \uparrow$ or $\downarrow$ incident on the cell boundary. Equation (23) implies that a $\uparrow$-electron arriving at the N cell at time $t'$ induces a transient variation $p(t - t')$ of the N-cell spin. Therefore, the function $p(t)$ has a width on the order of $\tau^{N}$, the mean time it takes an electron to traverse the interfacial metal layer. While $\langle i_{\sigma}^{in}\rangle = 0$, the fluctuations of the current can be modeled by the well-known result[42]

$$\langle i_{\sigma}^{in}(t)i_{\sigma'}^{in}(t')\rangle \propto \delta_{\sigma\sigma'}\delta(t - t') \sum_{k:\nu_{k,z}<0} n_{k}(t)[1 - n_{k}(t)]\nu_{k,z}, \tag{24}$$

where $\nu_{k,z}$ is the $z$ component of the group velocity of Bloch state $k$. We assume constant $\nu_{k,z}$ and isotropic electronic occupation numbers $n_{k}(t) = n(\epsilon_{k}, t)$, where $\epsilon_{k}$ denotes the band structure. Consequently, Eq. (24) simplifies to

$$\langle i_{\sigma}^{in}(t)i_{\sigma'}^{in}(t')\rangle \propto \delta_{\sigma\sigma'}\delta(t - t')k_{B}\tilde{T}_{e}^{N}(t)D(\epsilon_{F}) \tag{25}$$

where $\tilde{T}_{e}^{N}(t)$ is given by Eq. (3) of the main text.

Comparison of Eqs. (23) to (15) and (25) to (14) reveals the remarkable correspondence $i_{\uparrow}^{in} - i_{\downarrow}^{in} \leftrightarrow r^{N}$, $p \leftrightarrow \chi^{N}$ and $\tilde{T}_{e}^{N}(t) \leftrightarrow T^{N}$ between our spin-noise model and Langevin theory. The analogy identifies the orbital degrees of freedom of the N electrons as the bath that is coupled to the N-cell spins. In addition, as discussed in the context of Eq. (3), $\tilde{T}_{e}^{N}(t)$ can be interpreted as a generalized electron temperature. We are, therefore, led to set $T^{N}(t) = \tilde{T}_{e}^{N}(t)$ to good approximation in Eq. (16), thereby extending this fluctuation–dissipation relationship of the N layer to non-Fermi–Dirac electron distributions.

We note that the definition of a generalized temperature of a nonthermal electron system is not unique and depends on the considered property. The $\tilde{T}_{e}^{N}$ introduced here quantifies the noise intensity of the stream of conduction electrons incident on the F|N interface. It does in general not simply scale with the total excess energy density of the electrons, which was used previously to define a generalized temperature[47].

Similar considerations can be applied to the correlation function of the F-cell spin, if one wishes to go beyond the Langevin-type result of Eq. (16). According to Eq. (29) and ref.[11], the spin current $j_{s}^{F}(t)$ due to F-cell spin fluctuations can be expressed by the occupation numbers of all magnon modes, including nonthermal populations. However, since in our experiment the pump-induced changes of the YIG layer are negligible, we do not consider this aspect further.

**Calculation of $\kappa^{N}$ and $\kappa^{F}$.** Our numerical calculations are based on Eqs. (21) and (22). For the N layer, we assume

$$\chi^{N}(t) = \chi_{DC}^{N}\Theta(t)\frac{\exp(-t/\tau^{N})}{\tau^{N}}, \tag{26}$$

where $\tau^{N} \sim 4$ and 1 fs, respectively, is determined by using the Fermi velocity of Pt (ref.[77]) and Cu (ref.[78]). The N-cell DC spin susceptibility $\chi_{DC}^{N}$ is related to the paramagnetic susceptibility[43] $\tilde{\chi}_{DC}^{N}$ of Pt (ref.[44]) and Cu (ref.[79]) through $\chi_{DC}^{N} = a^{3}\tilde{\chi}_{DC}^{N}/\mu_{0}g^{2}\mu_{B}^{2}$, where $\mu_{0}$ is the vacuum permeability, $g = 2$, and $\mu_{B}$ is the Bohr magneton. The factor $a^{3}$ is required since $\chi_{0}^{N}$ refers to the integrated N-cell volume whereas $\tilde{\chi}_{0}^{N}$ is given per volume. The factor $\mu_{0}g^{2}\mu_{B}^{2}$ accounts for the different units used in the definition of $\tilde{\chi}_{0}^{N}$ and $\chi_{0}^{N}$.

For F=YIG, we determine the $\chi^{F}$ tensor by the Kubo formula (Eq. (13)) using the equilibrium spin correlation functions $\langle s_{j}^{F}(t)s_{l}^{F}(t')\rangle = \langle s_{j}^{F}(t - t')s_{l}^{F}(0)\rangle$ with $t > t'$ as an input. These functions were calculated by atomistic spin-dynamics simulations[12,45,76] in which ~$10^{6}$ $Fe^{3+}$ spins were propagated classically according to the YIG spin Hamiltonian plus a thermal noise field provided by a thermostat with temperature 300 K. Trajectories $\mathbf{s}^{F}(t)$ were obtained by summing all 20 $Fe^{3+}$ spins of a selected YIG unit cell. Note that this summation is approximately tantamount to summing up magnon amplitudes over all wavevectors and magnon branches[12]. The ensemble average was obtained by averaging the product $s_{j}^{F}(t - t')s_{l}^{F}(0)$ over many trajectories.

**Estimate of the SSE coefficient.** As a cross check, we use Eq. (21) to estimate the order of magnitude of $\mathcal{K} = \int dt \kappa^{N}(t)$. This formula can be simplified using Eq. (17) and yields the spin Seebeck coefficient

$$\mathcal{K} = \frac{k_{B}J_{sd}^{2}}{a^{2}} \int dt \chi^{N}(t) \cdot \left(\Theta * \chi_{\perp}^{F}\right)(t), \tag{27}$$

where $\chi_{\perp}^{F}(t) = \chi_{yz}^{F}(t) - \chi_{zy}^{F}(t)$. As $\chi^{N}(t)$ is localized around $t = 0$, we approximate $\left(\Theta * \chi_{\perp}^{F}\right)(t) \approx \chi_{\perp}^{F}(t = 0^{+})t$, use Eq. (26) and obtain

$$\mathcal{K} = \frac{k_{B}J_{sd}^{2}\chi_{\perp}^{F}(0^{+})\chi_{DC}^{N}\tau^{N}}{a^{2}} = \frac{2k_{B}J_{sd}^{3}\left|\langle S^{F}\rangle\right|\chi_{DC}^{N}\tau^{N}}{\hbar a^{2}}. \tag{28}$$

In the last step, we estimated $\chi_{\perp}^{F}(0^{+})$ by solving the equation of motion $\hbar\Delta\dot{\mathbf{s}}^{F} \approx \langle \mathbf{S}^{F}\rangle \times J_{sd}\mathbf{s}^{N}$ and comparison to $\Delta\dot{\mathbf{s}}^{F} = \underline{\chi}^{F} * J_{sd}\mathbf{s}^{N}$. We find $\chi_{\perp}^{F}(t) = \Theta(t) \cdot 2\left|\langle S^{F}\rangle\right|/\hbar$ for times $t \approx 0$, where $\left|\langle S^{F}\rangle\right| \approx 7$ is the total spin of the YIG unit cell at room temperature. Consideration of Eqs. (5) and (28) and the peak of the measured $j_{s}(t)$ (Fig. 5g) yields the estimate $J_{sd} \sim 2$ meV.

Equation (28) also allows us to compare the spin Seebeck coefficient of YIG|Pt and YIG|Cu|Pt. Assuming the $J_{sd}$ is the same for YIG|Pt and YIG|Cu interfaces and using Eq. (28), we find that $\mathcal{K}$ of YIG|Cu is a factor of about 2 smaller than that of YIG|Pt because of the different spin susceptibility $\chi^{N}(t)$ (see above). This difference in $\mathcal{K}$ provides a further reason for our observation that the YIG|Cu|Pt sample delivers a factor of 6 smaller THz emission signal than YIG|Pt (see Fig. 2e).

**Spin-mixing conductance.** Our equations for the SSE current are formulated in terms of the constant $J_{sd}$ that quantifies the coupling strength of electron spins at the F|N interface[3]. To connect to works[54,55] that formulate the SSE in terms of the spin-mixing conductance $g^{\uparrow\downarrow}$, we consider the current $j_{s}^{F}$ arising from the fluctuations of the F spins. Assuming an isotropic susceptibility of the N-cell spins and approximating $\mathbf{s}^{F}(t')$ by $\mathbf{s}^{F}(t) + \dot{\mathbf{s}}^{F}(t)(t' - t)$ in Eq. (11) yields

$$\mathbf{j}_{s}^{F}(t) = \langle \mathbf{s}^{F}(t) \times \dot{\mathbf{s}}^{F}(t)\rangle\frac{J_{sd}^{2}}{a^{2}} \int dt' \chi^{N}(t')t'. \tag{29}$$

This relationship agrees with the familiar result for thermal spin pumping, which is usually written as $\langle \mathbf{s}^F(t) \times \dot{\mathbf{s}}^F(t) \rangle \hbar \, \mathrm{Re}\, g^{\uparrow\downarrow}/4\pi \left| \langle \mathbf{S}^F \rangle \right|^2$ (ref.[37]). Comparison of the prefactors in both equations and use of Eq. (26) yields

$$\mathrm{Re}\, g^{\uparrow\downarrow} = \frac{4\pi \left| \langle \mathbf{S}^F \rangle \right|^2 J_{sd}^2 \chi_{DC}^N \tau^N}{\hbar a^2} \qquad (30)$$

and, thus, $\mathrm{Re}\, g^{\uparrow\downarrow} \sim 1 \times 10^{18}$ m$^{-2}$, in good agreement with calculated[38] and measured values[54,55]. Material parameters relevant for the calculations and estimates are summarized in Supplementary Table 1.

**Electron-dynamics simulations**. For a realistic simulation of the evolution of the electron distribution function $n(\epsilon, t)$ of a given material, we make use of the Boltzmann equation. Optical excitation, electron–electron and electron–phonon scattering are explicitly modeled by collision integrals[46,47]. All integrals take the density of states and quantum statistics of electrons and phonons of the material under study into account. For the screened electron–electron and electron–lattice Coulomb interaction, the screening parameter is calculated based on the instantaneous electron distribution function.

Instead of considering the electronic band structure of the material over the complete wavevector space, we introduce an effective one-band model in which an averaged isotropic dispersion relation is derived from the density of states $D(\epsilon)$. By tightly discretizing the energy space, we obtain a system of about 2700 coupled integro-differential equations which was numerically propagated in time. Material parameters relevant for the simulations are summarized in Supplementary Table 2.

**Data availability**. The data that support the findings of this study are available from the corresponding author upon reasonable request.

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

## Acknowledgements

We thank G.E.W. Bauer for stimulating discussions and acknowledge funding by the ERC H2020 CoG project TERAMAG/Grant No. 681917, the collaborative research center SFB TRR 227 Ultrafast spin dynamics (projects B01, B02 and B03), the collaborative research center SFB TRR 173 Spin+X (projects A01, A08 and B02), the Marie Curie FP7 project ITN WALL/Grant No. 608031, the DAAD (SpinNet, MaHoJeRo) as well as the DFG priority programs SPP 1538 SpinCaT and SPP 1666 Topological Insulators.

## Author contributions

T.S.S. and T.K. conceived the experiments. S.J., J.C., S.W.,C.C., G.J. and M.K. fabricated and characterized the samples. T.S.S. carried out the terahertz experiments and optical sample characterization with support from O.G. The transient reflectance measurements were performed by I.R. and A.M. Experimental data were analyzed by T.S.S. and T.K. with contributions from A.M. The theoretical model was developed by T.K. and P.W.B. with contributions from T.S.S. and J.B. Atomistic spin-dynamics simulations of YIG were conducted by J.B. Electron-dynamics simulations of Pt were conducted by S.T.W. and B.R. The manuscript was written by T.K. and T.S.S., with contributions from J.B., I.R., J.C., S.F.M., L.N., A.M., G.J., M.M., S.T.B.G., G.W., B.R., P.W.B., M.W. and M.K. All authors contributed to discussing the results and writing the paper.

## Additional information

**Competing interests:** The authors declare no competing Interests.

