## [Peer Review File · Nature Communications]

Reviewers' Comments:

Reviewer #1 (Remarks to the Author):

The authors describe their experiments and theoretical modeling of the spin Seebeck effect at Pt/YIG interfaces. The generation of spin currents by temperature differences at Pt/YIG interfaces (and temperature gradients near Pt/YIG interfaces) has been extensively reported in recent years. The authors have significantly advanced understanding by improving the time-resolution of these studies by more than an order-of-magnitude over what was achieved in Ref. 20. The authors make a convincing argument that a key parameter of the physics is the rate at which hot electrons internally thermalize to generate a high temperature distribution of electron-hole pairs; and that the observed time-response of the electric-fields generated by the inverse spin Hall effect is predominately governed by that time scale. I highly recommend publication in Nature Communications after the authors have considered the following comments. (To the editor: I did not attempt to check each step of the theoretical derivation of the spin current that is described across 5 pages of the methods section.)

- 1) The title, “Launching magnons at terahertz speed”, reads more like the title of a news release than the title of scholarly article. Speed typically has units of m/s and therefore terahertz speed is not a technically consistent statement. The word “launching” implies velocity is imparted but the velocity of magnons was not studied in these experiments. When a photon creates an electron-hole pair we do not say that the electron-hole pair is “launched” by the photon.
- 2) A minor point: I recommend that the authors state how much chirp GGG substrate produces. In other words, what is the GDD of GGG? (The pump pulse passes through the GGG substrate).
- 3) I recommend that the authors highlight a discussion of the temperature dependence reported in Fig 3b (the quadratic exponent) in the context of the theory developed in the methods section. Is this exponent understood?
- 4) Some experimental details and modeling parameters are missing or incomplete. The preparation of the YIG surface prior to Pt deposition is extremely important in this type of experiment. In other words, the spin conductance is known to depend strongly on the preparation conditions. The authors should describe the details of how the YIG surface is cleaned or processed prior to Pt deposition. Similarly, the authors should state, maybe in a table, the parameters that are inputs to the modeling and what prior work or current experiment was used to determine those parameters. For example, the electron-phonon coupling constant in Pt is often taken from Ref. 40 but some recent work has found a coupling parameter that is half as big as Ref. 40. The electronic heat capacity of Pt is usually taken from the linear-in-temperature heat

capacity measured at low temperatures but it is known that the heat capacity near room temperature is smaller than an extrapolation of the cryogenic results to room temperature would suggest.

5) The authors are correct to place their work in the context of prior work using photoemission to probe the electron energy distribution (Ref. 39). The laser fluence used in Ref. 39 however is an order of magnitude larger than what is used in the current experiment. Also, my understanding is that the internal thermalization of electrons at energies far from the Fermi energy should happen on the time scale of the inverse plasmon frequency. In a d-band metal, there are also scattering processes that should rapidly create a thermal distribution within the electron system. In other words, I recommend the authors provide more discussion and justification that the relatively long time-scale observed for the onset of spin transfer is really what is expected for the internal thermalization time of the electrons.

Reviewer #2 (Remarks to the Author):

The manuscript reported an interesting study on ultrafast spin Seebeck effect with unprecedented time resolution of 10 femtosecond, which allowed the authors to, for the first time, reveal the ultrafast dynamics of spin Seebeck effect. The sample studied is a bilayer structure of YIG/Pt. A 10-fs laser pulse from a Ti: sapphire laser was used to excite electrons in Pt and consequently created, on ultrafast time scale, a temperature difference between the insulating YIG layer and the metal layer. The magnetization of YIG, which is a ferromagnetic insulator, drives a spin current from YIG to Pt via spin Seebeck effect. Due to the inverse spin Hall effect, the spin current in Pt induced a transverse charge current that oscillates at THz. The AC charge current radiates a THz wave, which is time resolved by electro-optic sampling by mixing the THz with another ultrafast laser pulse.

The physics processes involved in the measurement are rather complicated. As such, great care was taken by the authors to study characteristics of the detected signal as many experimental parameters. It appears that the authors made a convincing case that the signal is indeed indicative of the spin Seebeck effect. The data are of high quality. Although there have been a number of studies on dynamical spin Seebeck effect, this work represents a significant progress toward understanding the mechanisms behind spin Seebeck effects in such systems. It also reveals other related ultrafast dynamics of electrons. Overall, this is a very interesting work that is novel and of general interest in both spintronic and ultrafast communities. The manuscript was well-written.

However, the manuscript could be potentially improved if the authors could provide more discussion on the frequency of THz signal: what determines the frequency of the emitted THz signal. It should be determined by the transverse charge current due to inverse spin Hall effect.

What is the origin for the charge current to be an alternative current? A discussion on this process would help readers understand the work better.

Reviewer #3 (Remarks to the Author):

The article by Y. Seifert et al. addresses the emerging field of THz spin-electronics. There are already few publications where the relevance of this field for the understanding of the short-time-scale spin-spin interaction has been shown.

The present article demonstrate that YIG/Pt bilayers generate a THz pulse when excited using a femto-second laser pulse. The experiment is well conducted and the results summarized in Fig. 2 and Fig.3 have the expected behavior. The toy model depicted in Fig 4 b,c,d,e is reminiscent of the spin Hall magnetoresistance model. Thereafter a more quantitative description based on s-d exchange model is developed.

The main claim of the paper is that spin current generation occurs on a very short time scale (100 fs) and that it is correlated to the thermalisation of the photo-excited electrons in the metallic layer; a behavior that is well reproduced by the theoretical model.

This model is based on a coherent description on how the electrons spins interact with the magnetization. Few points need however to be addressed.

1) The authors have calculated the time domain transverse spin susceptibility of YIG. There is however also the longitudinal spin susceptibility that can give rise to a spin current J_s . The authors should comment on why they have ignored it in the discussion.

2) Related to point (1) : SSE is an incoherent effect i.e. to generate a spin current one needs only to induce magnetization dynamics through (for example) a thermal torque as depicted in the article. There is however no need for the Ferromagnetic moment to interact with the same electron that has induced the torque. Clarifying this point and its implications on the model would make the article more accessible to a larger audience.

Author points:

- The title is not adequate using the word “launching” is misleading: there is no propagation of the excited magnons, they are incoherent thermal magnons.
- It is not so clear in Figure S1 that the S(-) amplitude is zero in the parallel to M configuration. A comment would be welcome.

Preliminary remarks to all reviewers/list of changes

We would like to thank all reviewers for their interest in our work, the time they spent on our manuscript and their detailed, useful and encouraging comments that have led to a significantly improved paper.

Our response and manuscript changes according to the reviewers' suggestion are detailed further below. **In the new manuscript, text changes are highlighted yellow**, often followed by a code such as "[R2.C3]", meaning that the preceding text addresses comment#3 of reviewer#2. For completeness, we also provide a document in which the changes are not highlighted.

Let us just briefly summarize here the major points and changes.

Manuscript text

- We added a new panel (Fig4a) to better explain our procedure to extract the spin current $j_s(t)$ from the THz emission signal $S(t)$.
- We restructured the results part to make experimental data as self-explaining as possible. Our theoretical model is now introduced as late as possible. First, after the spin-current extraction, we directly compare dynamics of $j_s(t)$ to transient reflectance $DR(t)$ of YIG|Pt, which is a measure of the decay of the electronic temperature. The decay of $j_s(t)$ and $DR(t)$ has very similar time constant, which shows that j_s follows the decay of the electron temperature quasi-instantaneously on a time scale of 100fs. We accordingly added a new section "Transient reflectance" and $DR(t)$ data to the main text figure 4.)
- Second, to explain this ultrafast SSE response and the rise of the SSE current, we only then proceed with introducing the model, showing that $j_s(t)$ scales with an instantaneous generalized electron temperature (Eq. (5)). To provide evidence for this statement, we conducted simulations of the electron dynamics of photoexcited Pt, which also provide the dynamics of the generalized electron temperature. We find good agreement with the dynamics of $j_s(t)$ (please see our new Fig.5g), thereby supporting our interpretation.
- We added a section to the discussion part in which the temperature dependence of the SSE current amplitude is discussed.

Methods Section

- We provided more details on the current-extraction procedure and the eventual time resolution of the extracted spin current $j_s(t)$.
- We rewrote the section "Nonthermal electron distributions" in which the introduction of a generalized electron temperature is explained in a substantially more readable and elegant manner.
- We added details on the simulations of the electron dynamics in photoexcited Pt.

Supplementary Material

- We added new measurements of the polarization state of the THz signal $S(t)$ with an improved measurement protocol.
- We added results on the dependence of the THz signal $S(t)$ on the pump polarization (linear and circular).
- We added a calculation of the generalized temperature \tilde{T}_N^e vs temperature T_N^e of a Fermi-Dirac distribution.
- We added a figure outlining the current-extraction procedure of $j_s(t)$.
- We added two tables providing details on the inputs used for the numerical estimates and the calculations of the electron dynamics in Pt.

Reviewer #1:

The authors describe their experiments and theoretical modeling of the spin Seebeck effect at Pt/YIG interferences. The generation of spin currents by temperature differences at Pt/YIG interfaces (and temperature gradients near Pt/YIG interfaces) has been extensively reported in recent years. The authors have significantly advanced understanding by improving the time-resolution of these studies by more than an order-of-magnitude over what was achieved in Ref. 20. The authors make a convincing argument that a key parameter of the physics is the rate at which hot electrons internally thermalize to generate a high temperature distribution of electron-hole pairs; and that the observed time-response of the electric-fields generated by the inverse spin Hall effect is predominately governed by that time scale. I highly recommend publication in Nature Communications after the authors have considered the following comments.

Response: We would like to thank the reviewer for her/his interest in our work and her/his detailed, useful, and encouraging comments that have led to a significantly improved manuscript.

Reviewer#1 comment#0: To the editor: I did not attempt to check each step of the theoretical derivation of the spin current that is described across 5 pages of the methods section.

Response: We checked the derivation once more again and did not detect issues.

Action: To make the derivation more readable and better work out the symmetry of F and N layers, we applied the following changes in variable names: \mathbf{S} to \mathbf{S}^N , \mathbf{M} to \mathbf{S}^F . As commonly implemented, \mathbf{M} is now used as symbol for the macroscopic magnetization of F. Accordingly, the fracture-type \mathcal{M} is not necessary any more.

Reviewer#1 comment#1: 1) The title, "Launching magnons at terahertz speed", reads more like the title of a news release than the title of scholarly article. Speed typically has units of m/s and therefore terahertz speed is not a technically consistent statement. The word "launching" implies velocity is imparted but the velocity of magnons was not studied in these experiments. When a photon creates an electron-hole pair we do not say that the electron-hole pair is "launched" by the photon.

Response: We agree---the title should reflect the content of the work more precisely.

Action: We accordingly changed the title to the more precise "Femtosecond formation dynamics of the spin Seebeck effect revealed by terahertz spectroscopy".

Reviewer#1 comment#2: 2) A minor point: I recommend that the authors state how much chirp GGG substrate produces. In other words, what is the GDD of GGG? (The pump pulse passes through the GGG substrate).

Response: (i) We thank the reviewer for bringing up this important point as the time resolution is crucial for our experiment. We calculated the dispersion introduced by the 0.5-mm-thick GGG substrate based on the known refractive index data [Wood, Darwin L., and Kurt Nassau. "Optical properties of gadolinium gallium garnet." *Applied optics* 29.25 (1990): 3704-3707.] and found a GDD of 82 fs². Assuming the pulse incident on the GGG window is bandwidth-limited, we obtain a pulse duration of 26 fs after the GGG.

We emphasize, however, that in our experiment, we precompensate for the dispersion introduced by the GGG substrate to the largest extent by a pair of chirped mirrors and two quartz wedge. By tuning the total quartz-wedge thickness, we minimized the pulse duration at the sample position by a 2PA-photodiode. This was done with an additional 0.5 mm of BK7 glass in the beam, which then was replaced by the GGG substrate. According to GDD of BK7 and GGG, we calculate a pulse duration of 19 fs after the GGG.

(ii) We note that the ultimate time resolution of interest is the time resolution of the measured and extracted charge current flowing inside the sample. To estimate an upper limit of this time resolution from our experimental data, we present in the figure below a comparison of the SSE-related charge sheet current J . (which is odd in the sample magnetization) to the sheet-current component $J_+(t)$ (which is even in the sample magnetization).

$j_+(t)$ exhibits a sharp initial peak that can be fit by a Gaussian of FWHM 27 fs. This width coincides with the FWHM of the Fourier-transformed low-pass filter function we employ in the extraction procedure of the transient current. (At high frequencies, the signal-to-noise ratio is too small for a reliable inversion.) This suggests that the time resolution of the photocurrent (which is the ultimate quantity of interest) is not limited by the duration of the pump pulse. (In fact, we use chirped mirrors and wedges for dispersion management to achieve minimum pump duration at the sample position.)

Therefore, 27 fs represents the upper limit to the time resolution of the measured photocurrent.

a. Retrieved charge sheet currents, even and odd in the magnetization of the YIG|Pt sample. Curves are offset for clarity. b. The magnified version on the right-hand panel also shows a Gaussian fit with a FWHM of 27 fs.

Action: We accordingly added text to the main text (at "Ultrafast SSE current") and the Methods section and now in particular mention the GDD of the 0.5-mm-thick GGG and the time resolution of the retrieved charge current.

Reviewer#1 comment#3: 3) I recommend that the authors highlight a discussion of the temperature dependence reported in Fig 3b (the quadratic exponent) in the context of the theory developed in the methods section. Is this exponent understood?

Response: We thank the reviewer for this question. Within the framework of our modeling, we find that the temperature dependence of the THz-SSE-field amplitude must arise from the product $J_{sd}^2 \chi_{\perp}^F(O^+)$ of the exchange constant and the YIG transverse susceptibility at the interface (also compare to Eq. 28 of Methods section and new section in the discussion part of the main text). This notion is in line with the discussion of Uchida et al. (PRX2014) who ascribe the temperature dependence of the DC SSE voltage to that of the spin-mixing conductance.

Action: We added a new paragraph to the discussion section in which the temperature dependence of the THz SSE field amplitude is discussed in the framework of our modeling.

Reviewer#1 comment#4: 4) Some experimental details and modeling parameters are missing or incomplete. The preparation of the YIG surface prior to Pt deposition is extremely important in this type of experiment. In other words, the spin conductance is known to depend strongly on the preparation conditions. The authors should describe the details of how the YIG surface is cleaned or processed prior to Pt deposition.

Response: This is an excellent idea and we fully agree with the referee that the preparation details were shown in previous studies to have an impact on the DC SSE.

Action: We accordingly added details of the preparation to the Methods section "Sample preparation".

Reviewer#1 comment#4a: (i) Similarly, the authors should state, maybe in a table, the parameters that are inputs to the modeling and what prior work or current experiment was used to determine those parameters. For example, the electron-phonon coupling constant in Pt is often taken from Ref. 40 but some recent work has found a coupling parameter that is half as big as Ref. 40.

(ii) The electronic heat capacity of Pt is usually taken from the linear-in-temperature heat capacity measured at low temperatures but it is known that the heat capacity near room temperature is smaller than an extrapolation of the cryogenic results to room temperature would suggest.

Response: (i) We strongly agree---an additional table presenting the respective parameters will improve the accessibility significantly. We note that with our new calculations of the generalized temperature $\tilde{T}_N^e(t)$ (see Eq. 3 and Fig. 5g), two-temperature-model calculations are not required any more. This approach is justified by the good agreement between the generalized temperature and the actual temperature for Pt in the experimental temperature range (see Fig. S5).

Action: We added two tables to the Supplementary Material.

We added calculations of the generalized temperature $\tilde{T}_N^e(t)$ (new Fig. 5g).

Reviewer#1 comment#5: 5) (i) The authors are correct to place their work in the context of prior work using photoemission to probe the electron energy distribution (Ref. 39). The laser fluence used in Ref. 39 however is an order of magnitude larger than what is used in the current experiment.

(ii) Also, my understanding is that the internal thermalization of electrons at energies far from the Fermi energy should happen on the time scale of the inverse plasmon frequency. In a d-band metal, there are also scattering process that should rapidly create a thermal distribution within the electron system. In other words, I recommend the authors provide more discussion and justification that the relatively long time-scale observed for the onset of spin transfer is really what is expected for the internal thermalization time of the electrons.

Response: We strongly agree with the reviewer: our interpretation that the rise of the spin current is due to the electron thermalization should be bolstered with more arguments than ref39. (Please note that in this ref (Phys. Rev. B 66, 245420), the absorbed fluence was 1 mJ/cm^2 , but for a bulk crystal in which the energy is distributed over a larger depth as compared to our Pt thin film. Therefore, the deposited energy per volume is roughly comparable in both experiments.)

To provide more support for our interpretation, we conducted state-of-the-art simulations of the electron dynamics in laser-excited Pt based on the Boltzmann equation. These simulations also provide the dynamics of the generalized electron temperature as an output. We find good agreement with the dynamics of $j_s(t)$ (please see our new Fig.5g), thereby bolstering our interpretation.

As a further argument, we also refer to previous ultrafast photoemission work on Ru, another transition metal (see e.g. Lisowski, M., Loukakos, P.A., Bovensiepen, U., Stähler, J., Gahl, C. & Wolf, M. Ultra-fast dynamics of electron thermalization, cooling and transport effects in Ru(001), Applied Physics A 78, 165-176 (2004)). In this work, the number of photoinduced electrons above the Fermi energy and, thus, the generalized temperature increase $\Delta\tilde{T}_e^N(t)$ was observed to rise on a time scale of 100 fs for quite similar excitation densities (Fig. 8 in this reference).

We note that in the sp-type metal Al, in contrast, electron thermalization is known to proceed significantly faster because of the smaller Coulomb screening parameter and its larger electronic density of states (ref.43).

Action: We accordingly extended and restructured our argumentation, which is now more straightforward.

We added a figure with simulation results for the generalized temperature (Fig.5g).

We added text to the results section "Comparison to measured and simulated electron dynamics"

We added text to the first part of the "Discussion" section.

Reviewer #2:

The manuscript reported an interesting study on ultrafast spin Seebeck effect with unprecedented time resolution of 10 femtosecond, which allowed the authors to, for the first time, reveal the ultrafast dynamics of spin Seebeck effect. The sample studied is a bilayer structure of YIG/Pt. A 10-fs laser pulse from a Ti: sapphire laser was used to excite electrons in Pt and consequently created, on ultrafast time scale, a temperature difference between the insulating YIG layer and the metal layer. The magnetization of YIG, which is a ferromagnetic insulator, drives a spin current from YIG to Pt via spin Seebeck effect. Due to the inverse spin Hall effect, the spin current in Pt induced a transverse charge current that oscillates at THz. The AC charge current radiates a THz wave, which is time resolved by electro-optic sampling by mixing the THz with another ultrafast laser pulse.

The physics processes involved in the measurement are rather complicated. As such, great care was taken by the authors to study characteristics of the detected signal as many experimental parameters. It appears that the authors made a convincing case that the signal is indeed indicative of the spin Seebeck effect. The data are of high quality. Although there have been a number of studies on dynamical spin Seebeck effect, this work represents a significant progress toward understanding the mechanisms behind spin Seebeck effects in such systems. It also reveals other related ultrafast dynamics of electrons. Overall, this is a very interesting work that is novel and of general interests in both spintronic and ultrafast communities. The manuscript was well-written.

Response: We would like to thank the reviewer for her/his interest in our work and her/his detailed, useful, and encouraging comments that have led to a significantly improved manuscript.

Reviewer#2 comment#1: However, the manuscript could be potentially improved if the authors could provide more discussion on the frequency of THz signal: what determines the frequency of the emitted THz signal. It should be determined by the transverse charge current due to inverse spin Hall effect. What is the origin for the charge current to be an alternative current? A discussion on this process would help readers understand the work better.

Response: We strongly agree with the reviewer that it is important to better work out the relationship between the temporal/spectral shape of the actual THz signal (as measured by EO sampling) and that of the THz current flowing inside the sample. In particular, it should be explained how a unipolar current (as in Fig. 4b) leads to a bipolar THz signal (Fig. 4a).

Generally, the measured electro-optic signal $S(t)$ is related to the THz electric field $E(t)$ directly behind the sample by the convolution $S(t) = (h * E)(t) = \int dt' h(t - t')E(t')$. Here, the transfer function $h(t)$ accounts for propagation to the detection unit as well as the detector response function of the electrooptic-sampling process. We have determined this function by using an appropriate reference emitter.

The measured transfer function (inset of Fig 3a) exhibits a sharp bipolar feature around, which upon convolution with $E(t)$ approximately yields a signal proportional to the derivative of the field, $S(t) \propto \partial E / \partial t$. Equation (6) is inverted directly in the time domain by recasting it in the form of a matrix equation. As expected, the bipolar $S(t)$ approximately scales with the derivative of the unipolar $E(t)$. Note that DC component of $h(t)$ is zero because DC electric fields cannot propagate away from their source. We determine the missing DC component by using the causality principle: the pump-induced charge current inside the sample and, thus, $E(t)$ is zero before arrival of the pump pulse at $t = 0$.

We finally note that our measurement window is limited to 5 ps. This sets a lower bound of about 0.2 THz to the detectable frequency components. This means, that any current component slower than this (such as presumably the charge back-flow) is not detected in our setup.)

Action: We accordingly added text to the main text (subsection "Ultrafast spin Seebeck current") and the Methods section "Extraction of the THz current" in which the extraction of $j_c(t)$ from $S(t)$ is now explained and rationalized in detail.

We also added the raw data $S(t)$ (Fig. 4a) underlying the extracted current (Fig. 4b). The transfer function is shown in the inset of Fig. 4a.

Reviewer #3:

The article by Y. Seifert et al. addresses the emerging field of THz spin-electronics. There are already few publications where the relevance of this field for the understanding of the short-time-scale spin-spin interaction has been shown.

The present article demonstrate that YIG/Pt bilayers generate a THz pulse when excited using a femtosecond laser pulse. The experiment is well conducted and the results summarized in Fig. 2 and Fig.3 have the expected behavior. The toy model depicted in Fig 4 b,c,d,e is reminiscent of the spin Hall magnetoresistance model. Thereafter a more quantitative description based on s-d exchange model is developed.

The main claim of the paper is that spin current generation occurs on a very short time scale (100 fs) and that it is correlated to the thermalisation of the photo-excited electrons in the metallic layer; a behavior that is well reproduced by the theoretical model. This model is based on a coherent description on how the electrons spins interact with the magnetization. Few points need however to be addressed.

Response: We would like to thank the reviewer for her/his interest in our work and her/his detailed, useful, and encouraging comments that have led to a significantly improved manuscript.

Reviewer#3 comment#1: 1) The authors have calculated the time domain transverse spin susceptibility of YIG. There is however also the longitudinal spin susceptibility that can give rise to a spin current J_s . The authors should comment on why they have ignored it in the discussion.

Response: We thank the reviewer for pointing out the interesting aspect that the SSE current due to the fluctuations in Pt does not depend on the longitudinal spin susceptibility χ_{jj}^F of YIG.

From a formal viewpoint, our model surely allows for the possibility of a longitudinal susceptibility of the F layer (χ_{jj}^F is free and allowed to be nonzero). In the derivation of the spin current up to second order in J_{sd} , we find that the χ_{jj}^F do not contribute. As indicated by the derivation step from Eq(19) to Eq(21), this conclusion arises because the direction of the spins in Pt is distributed isotropically.

Intuitively, this result can be understood since spin fluctuations along different coordinate axes are uncorrelated in the isotropic N layer. For example, the first interaction of $s_j^N(t')$ with the F layer would induce a change $\propto \chi_{jj}^F(t-t')s_j^N(t')$ in the F-cell spin, which is parallel to the j axis. Because $\langle s_j^N(t)s_i^N(t') \rangle \propto \delta_{ji}$, the only relevant second interaction is due to $s_j^N(t)$, again along the j axis. Therefore, no torque results, and the longitudinal χ_{jj}^F does not contribute to κ^N . This cancellation does not occur for κ^F because spin fluctuations in F are correlated in different j directions.

Action: We accordingly added a remark to the Methods following Eqs(21,22).

Reviewer#3 comment#2: 2) Related to point (1) : SSE is an incoherent effect i.e. to generate a spin current one needs only to induce magnetization dynamics through (for example) a thermal torque as depicted in the article. There is however no need for the Ferromagnetic moment to interact with the same electron that has induced the torque. Clarifying this point and its implications on the model would make the article more accessible to a larger audience.

Response: We fully agree with the reviewer. In our modeling, $\mathbf{S}^N(t)$ is the spin of the whole N cell, not the spin of a single electron.

Action: We added text to the discussion section to emphasize explicitly that $\mathbf{S}^N(t)$ is the total spin of a N=Pt unit cell at the YIG/Pt interfaces.

Reviewer#3 comment#3: The title is not adequate using the word "launching" is misleading: there is no propagation of the excited magnons, they are incoherent thermal magnons.

Response: We fully agree---the title should reflect the content of the work more precisely.

Action: We accordingly changed the title to the more precise "Femtosecond formation dynamics of the spin Seebeck effect revealed by terahertz spectroscopy".

Reviewer#3 comment#4: It is not so clear in Figure S1 that the S(-) amplitude is zero in the parallel to M configuration. A comment would be welcome.

Response: The reviewer is absolutely right to ask about the contribution of the THz emission signal S(-) parallel to the sample magnetization. We ascribe this part to imperfections of the rotatable THz wire-grid polarizer. We repeated the experiment neglecting rotation of the polarizer but rotated the sample magnetization by 90° instead. The result of this improved measurement protocol is shown in Fig. S1 and demonstrates the linear polarization of the signal S(-) perpendicular to the sample magnetization.

Action: We repeated experiments with an improved measurement protocol and updated the data of Fig. S1.

Reviewers' Comments:

Reviewer #1 (Remarks to the Author):

The authors have adequately addressed the concerns raised in my first report and significantly improved their manuscript in their thorough responses to all of the reviewer comments. I highly recommend publication in Nature Communications.

Reviewer #2 (Remarks to the Author):

The authors have addressed my comments adequately. I also feel that the other reviewers have raised important questions in the first round of review, and in my opinion, the authors have made a strong effort to address these comments and improved the manuscript accordingly. I support publication of the revised manuscript on Nature Communications.

Reviewer #3 (Remarks to the Author):

The authors have given all required information to make this paper a constructive contribution to the field, I hence recommend its publication.